# Activity, structure, and diversity of Type II proline-rich antimicrobial peptides from insects

Weiping Huang [1,2,7], Chetana Baliga [1,2,6,7], Elena V Aleksandrova[3], Gemma Atkinson [4,5], Yury S Polikanov [1,2,3✉], Nora Vázquez-Laslop [1,2✉] & Alexander S Mankin [1,2✉]

## Abstract

**Apidaecin 1b (Api), the first characterized Type II Proline-rich antimicrobial peptide (PrAMP), is encoded in the honey bee genome. It inhibits bacterial growth by binding in the nascent peptide exit tunnel of the ribosome after the release of the completed protein and trapping the release factors. By genome mining, we have identified 71 PrAMPs encoded in insect genomes as pre-pro-polyproteins. Having chemically synthesized and tested the activity of 26 peptides, we demonstrate that despite significant sequence variation in the N-terminal sequence, the majority of the PrAMPs that retain the conserved C-terminal sequence of Api are able to trap the ribosome at the stop codons and induce stop codon readthrough—all hallmarks of Type II PrAMP mode of action. Some of the characterized PrAMPs exhibit superior antibacterial activity in comparison with Api. The newly solved crystallographic structures of the ribosome complexed with Api and with the more active peptide Fva1 from the stingless bee demonstrate the universal placement of the PrAMPs' C-terminal pharmacophore in the post-release ribosome despite variations in their N-terminal sequence.**

**Keywords** Ribosome; Antibiotics; PrAMP; Translation; Structure
**Subject Categories** Evolution & Ecology; Microbiology, Virology & Host Pathogen Interaction; Structural Biology

## Introduction

The spread of antibiotic resistance among bacterial pathogens makes the search for new antimicrobial agents an urgent task. Antimicrobial peptides (AMPs) could be a rich source of new antibiotics (Fox, 2013; Magana et al, 2020). AMPs are encoded in the genomes of many organisms and constitute an important component of their innate immune system (Hancock and Sahl, 2006; Magana et al, 2020; Yu et al, 2018). Most of the known AMPs permeabilize bacterial cell membranes causing lysis (Brogden,

2005). However, some AMPs, such as proline-rich antimicrobial peptides (PrAMPs) produced by arthropods and mammals, act upon intracellular targets. Due to their non-lytic nature, PrAMPs typically exhibit low toxicity towards eukaryotic cells, which positions them as promising antimicrobial agents (Fosgerau and Hoffmann, 2015; Scocchi et al, 2011). These short cationic peptides sneak into bacterial cells by hijacking specific peptide transporters (Krizsan et al, 2015; Mattiuzzo et al, 2007) and, once in the cytoplasm, they bind to the ribosome and inhibit protein synthesis (Graf and Wilson, 2019).

PrAMPs interfere with translation by invading the ribosomal nascent peptide exit tunnel (NPET), a narrow passageway in the large ribosomal subunit, through which the proteins newly synthesized in the peptidyl transferase center (PTC) leave the ribosome (Fig. 1A). Based on their mode of binding and action, PrAMPs are classified as Type I or Type II (Graf and Wilson, 2019; Polikanov et al, 2018). Most of the studied PrAMPs belong to Type I. They arrest the initiating ribosome at mRNA start codons by invading the PTC and impeding accommodation of the first elongator aminoacyl-tRNA (Gagnon et al, 2016; Seefeldt et al, 2016; Seefeldt et al, 2015; Weaver et al, 2019). In contrast, Type II PrAMPs bind not to the initiating but to the terminating ribosomes (Florin et al, 2017; Koller et al, 2023; Mangano et al, 2023). When a ribosome reaches a stop codon, it associates with one of the class 1 release factors (RF1 or RF2, depending on the stop codon), that promote detachment of the completed nascent protein from the tRNA. Once the released nascent protein leaves the ribosome, a Type II PrAMP diffuses up the NPET and traps the RF and tRNA in the PTC. As a result, the ribosome is arrested at the stop codon in a post-release state. Soon after exposure of the cell to Type II PrAMPs, most of the cellular RF1 and RF2 proteins become sequestered in the PrAMP-stalled complexes. However, because there is a large excess of ribosomes over RFs in bacterial cells, the remaining ribosomes continue translation and after reaching the end of the protein-coding sequence, they fail to release the completed polypeptides due to the lack of available RFs. Some of these ribosomes, stalled at the stop codon in a pre-release state and carrying unhydrolyzed peptidyl-tRNA, eventually bypass the stop codon by either erroneously accepting a near-cognate aminoacyl-tRNA or via frameshifting and synthesize aberrant proteins with C-terminal extensions (Mangano et al, 2020).

[1]Department of Pharmaceutical Sciences, University of Illinois at Chicago, Chicago, IL 60612, USA. [2]Center for Biomolecular Sciences, University of Illinois at Chicago, Chicago, IL 60607, USA. [3]Department of Biological Sciences, University of Illinois at Chicago, Chicago, IL 60607, USA. [4]Department of Experimental Medical Science, Lund University, 221 00 Lund, Sweden. [5]Virus Centre, Lund University, Lund, Sweden. [6]Present address: Department of Biotechnology, Faculty of Life and Allied Health Sciences, M.S. Ramaiah University of Applied Sciences, New BEL Road, MSR Nagar, Bangalore, Karnataka 560054, India. [7]These authors contributed equally: Weiping Huang, Chetana Baliga. ✉E-mail: yuryp@uic.edu; nvazquez@uic.edu; shura@uic.edu

Apidaecin, an AMP produced by the honey bee *Apis mellifera*, is the first discovered Type II PrAMP (Casteels et al, 1989; Florin et al, 2017). In the honey bee genome, apidaecin is encoded as a pre-pro-protein containing multiple repeats of nearly identical isoforms separated by short, conserved oligopeptide spacers used as a processing signal for carboxy-, amino-, and endopeptidases (Casteels-Josson et al, 1993). The major variant, apidaecin 1b (referred to throughout the rest of the paper as Api), is an unmodified 18 amino acid-long peptide with the sequence GNNRPVYIPQPRPPHPRL. Wild-type (wt) Api shows promising activity against a range of Gram-negative and some Gram-positive bacteria (Berthold et al, 2013; Casteels et al, 1989; Casteels et al, 1994; Czihal and Hoffmann 2009). Its derivatized variant, Api137, is characterized by a somewhat improved antibacterial activity and serum stability (Berthold et al, 2013). Cryogenic electron microscopy (cryo-EM) structural studies of the ribosome-Api137 complex show that it binds in the NPET in the same orientation as a growing polypeptide chain synthesized by the ribosome, with the peptide's C-terminus approaching the PTC active site and the N-terminus protruding down the tunnel (Fig. 1A) (Florin et al, 2017). The penultimate arginine (R17) of Api interacts with the glutamine side chain of the universally conserved GGQ motif of the RF, whereas the carboxyl group of Api's C-terminal leucine (L18) closely approaches the 3′ terminal ribose of the P-site tRNA. Specific contacts of several C-terminal residues of Api with rRNA and ribosomal proteins directs peptide's binding in the NPET positioning the PrAMP's 'warhead' (R17 and L18), for proper contacts with the RF and tRNA (Florin et al, 2017; Graf et al, 2018). Studies of chemically synthesized Api variants and screening of endogenously expressed *api* gene libraries show that the PrAMP's C-terminal segment is critical for the on-target activity and tolerates only a small number of mutations (Baliga et al, 2021; Berthold et al, 2013; Skowron et al, 2023; Taguchi et al, 2009; Taguchi et al, 1994; Taguchi et al, 1996). Even though single amino acid substitutions in the N-terminal segment usually do not abolish the ribosome-targeting activity of Api, some such mutations dramatically decrease the PrAMP's antibacterial potency, likely by interfering with the uptake or proteolytic stability of the peptide (Baliga et al, 2021).

Although testing Api variants produced via improvised chemical synthesis or identified by screening of random mutant gene libraries has been useful for establishing the basics of the structure-activity relationship, the majority of the variants identified by these intrinsically serendipitous approaches turned out to be either inactive or inferior to the natural prototype (Baliga et al, 2021; Berthold et al, 2013; Lai et al, 2019; Skowron et al, 2023; Taguchi et al, 2009; Taguchi et al, 1996). Native PrAMPs, in contrast, have been evolutionarily selected for their antibacterial activity over millions of years. In fact, even very early efforts carried out at the time when only scarce genomic data were available, revealed novel Api-like peptides encoded in insect genomes, and testing some of these PrAMPs showed their promising antibiotic activity (Casteels et al, 1989; Casteels et al, 1994; Li et al, 2006), but lack of adequate experimental tools made it impossible to verify either the cellular target or the mode of action of those PrAMPs. Thus, even thirty years after the initial discovery of Api, only a limited number of Type II PrAMPs have been found, verified, and investigated (Florin et al, 2017; Koller et al, 2023; Mangano et al, 2023). Identifying and testing new bioactive native peptides that could interfere with the

ribosome functions could offer a shortcut to finding antibacterials with properties superior to those of the known Type II PrAMPs.

In this study, by searching the significantly expanded genomic- and protein sequence databases we have identified a large variety of Api-like peptides encoded in insect genomes. By examining the mode of action of the newly discovered peptides we have found that the active ones arrest terminating ribosomes and induce stop codon readthrough, thereby exhibiting the mode of action that is characteristic for Type II PrAMPs. Notably, some of the new PrAMPs are more potent antibacterials than the prototype Type II PrAMP Api produced by honey bees. By solving the crystallographic structures of bacterial ribosomes in complex with Api, its derivative Api137, and one of the most active new PrAMPs, Fva1, we show that despite sequence differences in the N-terminal segments, pharmacophoric C-termini are placed in the ribosomal NPET in an invariable conformation that allows trapping of the RF and tRNA.

## Results

### Search for Api-like PrAMPs in sequenced genomes

In the genome of the honey bee *A. mellifera*, Api is encoded as a pre-pro-polyprotein with multiple repeats of Api isoforms separated by short spacers containing protease processing sites (Fig. 1B) (Casteels-Josson et al, 1993). In order to identify new Api-like PrAMPs, we used the amino acid sequence of both the active Api peptide and the Api precursor polyprotein, as well as the nucleotide sequence of the Api mRNA to search for homologs in the currently available protein and nucleotide databases. The combination of these approaches allowed us to identify many genes encoding Api-like PrAMPs. Although the searches were carried out across the global databases, new PrAMPs were found exclusively in insect genomes, specifically in the genomes of 45 species of bees and wasps. In the majority of cases, the Api-like peptides were encoded in pre-pro-polyproteins with a varying number of PrAMP repeats. Some genomes contained several PrAMP-encoding pre-pro-polyprotein genes, usually located on different chromosomes. Most commonly, the PrAMP sequences were separated from each other by a protease processing signal resembling the motif RREAEPEAEP found in the *A. mellifera* Api polyprotein (Casteels-Josson et al, 1993). The processing motifs were used as a guide to define the peptides' termini. However, due to the degenerate nature, variable length, and possible species-specificity of the processing motifs, the exact boundaries of the mature PrAMPs, especially at the N-termini, were sometimes ambiguous (Fig. 1B).

The majority of the PrAMP repeats in the identified polyproteins specified peptides with the characteristically conserved Api-like C-terminal sequence. Some polyproteins (e.g., the one encoded in the genome of *Vespula germanica*) contained repeats of identical PrAMPs (Fig. 1C), while others (e.g., the polyprotein encoded in the *Frieseomelitta varia* genome) encoded different Api-like peptide isoforms with sequence variations within the N-terminal peptide segment. However, in several species, e.g., *Bombus affinis*, the sequences of the PrAMPs in the polyprotein significantly deviated from each other: structures of some of the peptides closely resembled Api while others, although retaining the characteristic high Pro/Arg content of the PrAMP family, had

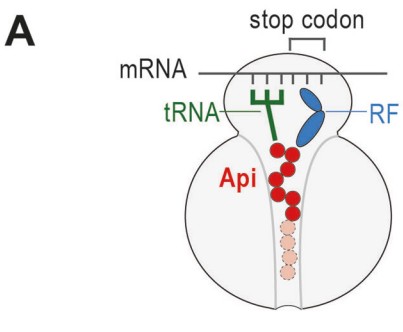

**A** stop codon

mRNA

tRNA    RF

Api

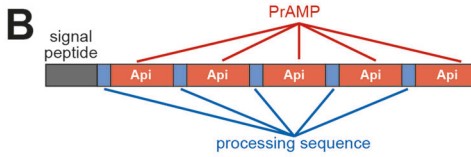

**B** PrAMP

signal peptide | Api | Api | Api | Api | Api

processing sequence

**C**

*Vespula germanica*

```
MKRCSLKASTINEPPCISNGLLNRNHWIQDLITKRLRREADPES        44
Vge1 ➤          NKPRPQQVPPRPPHPRLRREADPES            69
Vge1 ➤          NKPRPQQVPPRPPHPRLRREADPES            94
Vge1 ➤          NKPRPQQVPPRPPHPRLRREADPES           119
Vge1 ➤          NKPRPQQVPPRPPHPRLRREANPESSRI       147
```

*Frieseomelitta varia*

```
MKTFIFAILVVTFAVATCLDTMVTPSESLRLRREADPEPEP           41
Fva2 ➤   RNRPNIPKYIPPPRPPHPRLRREADPEPEP            71
Fva4 ➤   GNRAI---YVPPPRPPHPRLRREADPEPEP            98
Fva5 ➤   GNRPI---YVPPPRPPHPRLRREADPEPEP           125
Fva1 ➤   GNRPNIPTYIPRPRPPHPRLRREADPEPEP           155
Fva3 ➤   GSRPI---YVPPPRPPHPRLRREADPEPEP           182
Fva1 ➤   GNRPNIPTYIPRPRPPHPRLRREADPEPEP           212
Fva3 ➤   GSRPI---YVPPPRPPHPRLRREADPEPEP           239
Fva1 ➤   GNRPNIPTYIPRPRPPHPRLRREADPEPEP           269
Fva5 ➤   GNRPI---YVPPPRPPHPRLRREADPEPI           295
```

*Bombus affinis*

```
MKNFIFAILAITFVVAASTTVIPNAHKTELRRRREAGPEPEP          42
Baf3 ➤       SNRPPRPISLPPINPRLRREADPEPEP            69
Baf4 ➤       ANRPVYIPPPRPPHPRLRREADPEPEP            96
Baf4 ➤       ANRPVYIPPPRPPHPRLRREADPEPEP           123
Baf4 ➤       ANRPVYIPPPRPPHPRLRREADPEPEP           150
Baf4 ➤       ANRPVYIPPPRPPHPRLRREADPEPEP           177
Baf4 ➤       ANRPVYIPPPRPPHPRLRREADPEPEP           204
Baf4 ➤       ANRPVYIPPPRPPHPRLRREADPEPEV           231
Baf1 ➤       GSRLMYLPPPPCSPYLHFRREADPEAEV           259
Baf2 ➤       GYRSMYYPRPCPTHSHLRHGADTDAELYY           288
```

**D**

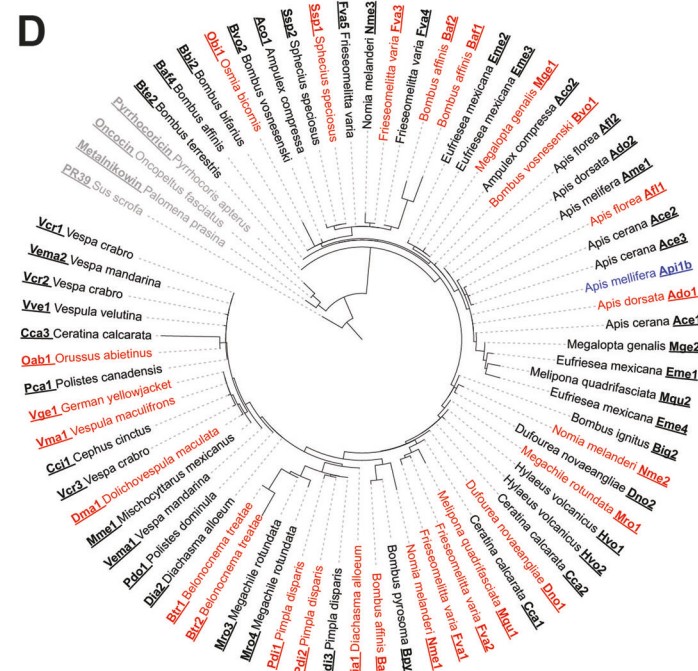

**E**

| | | |
|---|---|---|
| *Apis mellifera* | **Api** | GNNRPVYIPQPRPPHPRL |
| *Pimpla disparis* | **Pdi1*** | GKPNRPRPAPIQPRPPHPRL |
| *Pimpla disparis* | **Pdi2*** | GKPNKPRPAPIKPRPPHPRL |
| *Frieseomelitta varia* | **Fva1** | GNRPNIPTYIPRPRPPHPRL |
| *Frieseomelitta varia* | **Fva2** | RNRPNIPKYIPPPRPPHPRL |
| *Frieseomelitta varia* | **Fva3** | GSRPIYVPPPRPPHPRL |
| *Diachasma alloeum* | **Dia1** | SRPKPLPIPPRPPHPRL |
| *Dufourea novaeangliae* | **Dno1** | AKPSRPVYIPPPRPPHPRL |
| *Belonocnema treatae* | **Btr1** | GSKPVRPPPPIRPRPPHPRL |
| *Belonocnema treatae* | **Btr2** | ALPRPPPPFRPRPPHPRL |
| *Dolichovespula maculata* | **Dma1*** | GKPRPQQVPPRPPHPRL |
| *Vespula germanica* | **Vge1*** | NKPRPQQVPPRPPHPRL |
| *Megachile rotundata* | **Mro1** | ASRPVYVPPPRPPHPRL |
| *Megalopta genalis* | **Mge1** | ALNRPVYVPPPRPPHPRL |
| *Nomia melanderi* | **Nme2** | AHKPVYVPPPRPPHPRL |
| *Vespula maculifrons* | **Vma1*** | SNKPRPQQVPPRPPHPRL |
| *Sphecius speciosus* | **Ssp1*** | NRPTYVPPPRPPHPRL |
| *Orussus abietinus* | **Oab1** | SRPRPQQVPPPQPHPRL |
| *Osmia bicornis* | **Obi1** | GTKPLYIPRPPPQPHPRL |
| *Apis florea* | **Afl1** | GNNRPVYIPQPRPPHPRT |
| *Bombus vosnesenski* | **Bvo1** | SNRPVYIPPPRSPHPRL |
| *Bombus affinis* | Baf1 | GSRLMYLPPPPCSPYLHF |
| *Bombus affinis* | Baf2 | GYRSMYYPRPCPTHSHL |
| *Bombus affinis* | Baf3 | SNRPPRPISLPPINPRL |
| *Melipona quadrifasciata* | Mqu1 | DNRPDILIYIPPPGPPHTRL |
| *Apis dorsata* | Ado1 | GNNRPVYIPQPRPPHP |
| *Nomia melanderi* | Nme1 | MSITRPPPRPPHIRL |

**F**

basic  hydrophobic  neutral  polar

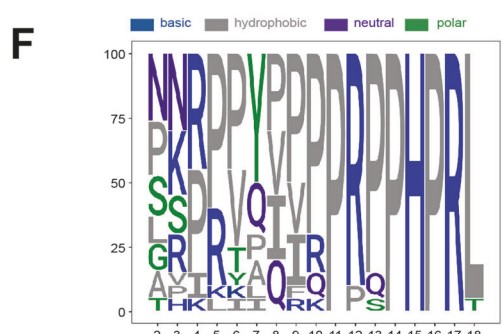

**Figure 1. Many Apidaecin-like PrAMPs are encoded in insect genomes.**

(A) The prototype Type II PrAMP Api stalls the ribosome at the stop codon, trapping the release factor (RF) and the deacylated tRNA. The functionally critical C-terminal segment of Api is shown in red. (B) The organization of the Apidaecin gene in the genome of the honey bee *A. mellifera*. (C) Examples of insect polyproteins with Api-like PrAMP repeats. The N-terminal leader sequence is shown in black, the PrAMP sequences are in red and the putative protease processing signals are in blue. The conserved C-terminal amino acid sequence is highlighted. A putative PrAMP with amino acid sequence significantly deviating from that of Api is shown in gray. Amino acid residue numbering is shown on the right and the assigned name of the corresponding PrAMP is indicated on the left. (D) Maximum likelihood tree showing sequence relationships between PrAMPs found in the polyproteins encoding Api-like peptides (branches with bootstrap support values greater than 50% are depicted as thick lines). Peptides that were chemically synthesized and experimentally tested are shown in red. Type I PrAMPs Metalnikowin, Oncocin, Pyrrhocoricin and cathelicidin-type PrAMP PR-39 (all shown in gray) were used as an outgroup. (E) PrAMPs that were chemically synthesized and experimentally tested. PrAMPs with the conserved Api-like C-terminal sequence are shown in red and the conserved sequence is highlighted. (F) Seqlogo analysis of the synthesized Api-like peptides shown in red in (E). The longer PrAMPs were N-terminally truncated to the length of 17 amino acid residues corresponding to the size of the shortest peptide in the list. Amino acid numbering corresponds to that in Api.

substantially different sequences (Fig. 1C). The different PrAMP variants within the same polyprotein may represent isoforms with similar activity (Casteels-Josson et al, 1993). Alternatively, a single polyprotein may encode an assortment of PrAMPs with different mechanisms of action and targets (Hanson et al, 2016; Hanson et al, 2022). It is also possible that some PrAMP-like sequences could be remnants of deteriorating inactive PrAMPs.

Altogether, we have identified 71 non-redundant PrAMPs that deviated from the 18 amino acid-long Api in lengths (ranging from 15 to 34 amino acids) and sequence (Table EV1). New PrAMPs were named using the 3-character designation of the species from which they are derived followed by the number indicating different peptide isoforms present in the polyprotein.

The global sequence alignment of the known and newly identified PrAMPs of the Api family shows significant conservation of the C-terminal 8-amino acid long sequence PRPPHPRL. In contrast, the N-terminal segments exhibit considerable sequence variation (Table EV1). These observations reinforce the notion of the functional importance of Api's C-terminal sequence, which emerged from the early comparison of several isolated Api-like PrAMPs (Casteels et al, 1994) as well as from the studies of Api mutants (Baliga et al, 2021; Berthold et al, 2013; Castle et al, 1999; Huang et al, 2024; Taguchi et al, 2009; Taguchi et al, 1994; Taguchi et al, 1996).

Due to the short length of the Api-like peptides and the conservation of the C-terminal sequence, the phylogenetic relationships between them are inevitably ambiguous. Nevertheless, as a guide for selection of the new PrAMPs for experimental testing, we generated a maximum likelihood tree (Fig. 1D) and selected 26 putative new PrAMPs for chemical synthesis and evaluation of their ribosome-targeting and antibacterial activity (Fig. 1E). Most of these peptides have the conserved Api-like C-terminal segment PRPPHPRL, but a highly diverse N-terminal sequence (Fig. 1E,F). However, we also synthesized and tested few peptides with amino acid variations in the conserved C-terminal segment as well as several PrAMPs co-encoded in the polyproteins with Api-like peptides but whose sequences significantly deviated from that of Api (Fig. 1E).

## Api-like PrAMPs exhibit antibacterial activity

For the initial evaluation of the antibacterial activity of the synthesized PrAMPs, we used a drop-diffusion test in which a small volume of peptide solution is placed on a lawn of bacterial cells growing on an agar plate. Lack of cell growth in the vicinity of the site of application reveals the antimicrobial activity of the tested

compound. In this assay, most of the PrAMPs inhibited growth of a laboratory *Escherichia coli* strain and of a multi-drug resistant clinical isolate of *Klebsiella pneumoniae* (Fig. 2). Notably, the majority of the PrAMPs that failed to inhibit *E. coli* or *K. pneumoniae* growth lacked the conserved PRPPHPRL C-terminal sequence.

The size of the clearing zone on the *E. coli* or *K. pneumoniae* lawns generated by some of the new PrAMPs, e.g., Dno1, Btr1, or Fva1, equaled or possibly even exceeded that produced by Api. However, assessing the antibacterial activity of PrAMPs based on the diameter of the clearing zone in drop-diffusion assay could be misleading because the diffusion through the agar-solidified media may be significantly affected by the physico-chemical properties of the peptide. Therefore, to gain a more accurate account of the antibacterial properties of the new PrAMPs, we determined the minimal inhibitory concentrations (MICs) of the most potent peptides in a liquid culture by microdilution assay. The results of the MIC testing in two different media confirmed that the activity of several of the new PrAMPs against *E. coli* or *K. pneumoniae* exceeded that of Api by 2- to 16-fold (Table 1). The most potent among the tested peptides were Pdi1, Pdi2, Fva1, Fva2, Dia1, Dno1, and Btr1. In summary, we found that the majority of the newly identified PrAMPs that retained the conserved Api-like C-terminal sequence readily inhibited growth of sensitive clinical and laboratory bacterial strains. Importantly, the activity of some of the new PrAMPs significantly exceeded that of Api.

### New Api-like PrAMPs arrest the terminating ribosome

The inability of some of the synthesized PrAMPs to interfere with bacterial growth could result from their poor cellular uptake. Alternatively, these peptides could be unable to act upon the ribosome inside the cell. To distinguish between these two contrasting scenarios, we tested the on-target activity of the synthesized PrAMPs in a cell-free translation system, which alleviates the necessity of the peptide uptake into the cell.

Due to their peculiar mode of action leading to ribosome arrest after the release of the completed protein, Type II PrAMPs only marginally interfere with the expression of a reporter protein in cell-free translation systems (Krizsan et al, 2014; Lauer et al, 2024; Mangano et al, 2023). Therefore, to directly evaluate the ribosome-targeting activity of the new PrAMPs we employed in vitro toeprinting analysis, an assay that identifies the sites of antibiotic-induced ribosome stalling on mRNA (Florin et al, 2017; Orelle et al, 2013) (Fig. 3). The results of the toeprinting experiments showed that, similar to Api and its derivative Api137, 20 out of the 26 new

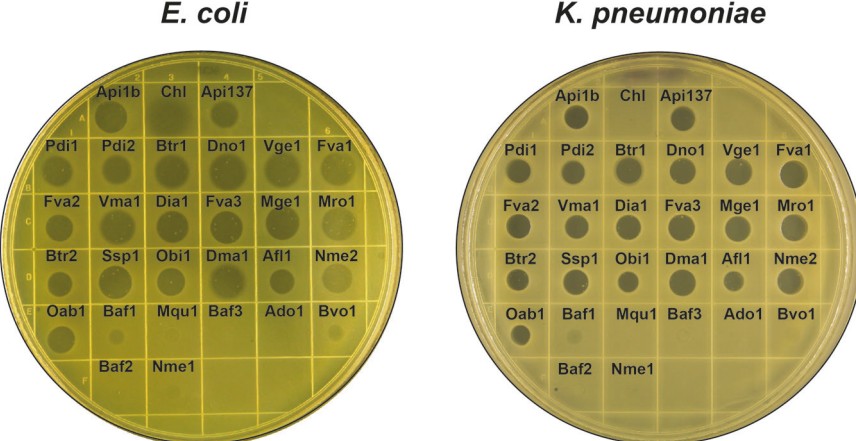

**Figure 2. Drop-diffusion test shows the antimicrobial activity of the synthesized PrAMPs.**

Individual synthetic peptides (2 µL drop of 2 mM solution) were spotted onto a lawn of bacterial cells growing on agar plates. Plates were imaged after 18 h incubation at 37 °C. The experiment was repeated twice and produced similar results.

PrAMPs arrested the ribosome at the stop codon, exhibiting the hallmark mode of action of Type II PrAMPs. Noteworthy, Bvo1, which had almost no antibacterial activity in the microbiological tests (Fig. 2; Table 1), readily arrested the ribosome at the stop codon (Fig. 3), indicating that its inability to inhibit bacterial growth stemmed from poor cellular uptake. Six of the new PrAMPs had no effect upon ribosome progression through mRNA (Fig. 3). These peptides are encoded in polyproteins together with Api-like PrAMPs, but their sequences significantly deviate from that of Api (Fig. 1E). We surmised that these non-Api-like peptides either require posttranslation modifications for their activity, are inactive specifically against the *E. coli* ribosome, have a different cellular target, or altogether have no antibacterial activity.

Overall, the in vitro toeprinting experiments demonstrate that the Api-like peptides, significantly differing from each other in their N-termini but maintaining the conserved C-terminal sequence PRPPHPRL, retain the ability to arrest the terminating ribosome. Because the tested peptides represent a variety of branches of the similarity tree (Fig. 1D), the results of the toeprinting analysis strongly argue that the majority of the Api-like peptides identified in insect genomes are biologically active and can stall ribosomes at stop codons.

## Api-like PrAMPs stimulate translational bypass of stop codons

Toeprinting analysis that showed PrAMP-induced ribosome stalling at the stop codon of the in vitro translated ORF does not fully reveal the mode of action of the inhibitor. Preventing RF binding or interfering with the peptidyl-tRNA hydrolysis would result in the appearance of the same toeprint signal as arresting the RFs on the post-release ribosome in a Type II PrAMP fashion. However, an important consequence of the latter mode of action is the rapid depletion of the available RFs in the cell and, as a result, stalling of the excess of active ribosomes at stop codons in a pre-release state. Many of such ribosomes, being unable to hydrolyze peptidyl-tRNA and liberate the completed proteins, eventually

bypass the stop codon by incorporating a near-cognate aminoacyl-tRNA (Baliga et al, 2021; Florin et al, 2017; Mangano et al, 2020; Mangano et al, 2023). Accordingly, we used an in vivo stop codon readthrough assay to verify that the newly identified antibacterial peptides present the genuine mode of action of Type II PrAMPs. To this end, we used the drop-diffusion assay with *E. coli* cells transformed with a reporter plasmid encoding the in-frame fused genes of red- and green fluorescent proteins (RFP and GFP, respectively), separated by a UGA stop codon (Mangano et al, 2023; Monk et al, 2017). Untreated cells express RFP but almost no GFP due to the termination of translation at the stop codon at the end of the *rfp* gene. However, induction of stop codon bypass should lead to the production of the hybrid protein exhibiting the characteristic GFP fluorescence (Mangano et al, 2023; Monk et al, 2017).

Applying drops of PrAMPs to the lawn of growing reporter cells produced a typical clearing zone of inhibition of cell growth, but in addition, cells growing at subinhibitory concentration of most of the tested Api-like PrAMPs generated a halo of GFP fluorescence revealing the PrAMP-induced stop codon readthrough (Fig. 4). These data, in conjunction with the results of the activity testing in a cell-free translation system, strongly argue that most of the new Api-like peptides are genuine Type II PrAMPs able to arrest the terminating ribosome and induce stop codon readthrough.

## The interactions of the C-terminal pharmacophore with the ribosome are preserved in the more potent PrAMP

Because the length and amino acid sequence of the N-terminal segment of some of the most active PrAMPs significantly deviated from that of Api, it was unclear whether these variations alter the interaction of the peptide's conserved C-terminal segment with the ribosome, P-site tRNA, and A-site RF. To address this question, we determined the X-ray crystal structures of the ribosome from the Gram-negative thermophilic bacterium *Thermus thermophilus* bound to native Api, its synthetic derivative Api137 and one of the most active new PrAMPs, Fva1, at 2.85, 2.70, and 2.70 Å resolution, respectively (Figs. 5 and EV1; Table 2). All the PrAMPs were co-crystallized with

**Table 1. Minimal inhibitory concentrations (MICs) of Api-like PrAMPs.**

| PrAMP | MIC (μM) | | | |
|---|---|---|---|---|
| | *E. coli* BL21 | | *K. pneumoniae* AR-0112 | |
| | 33% TSB | RPMI/Serum | 33% TSB | RPMI/Serum |
| **Apilb** | 0.5 | 4 | 4 | >64 |
| **Pdi1*** | 0.125 | 1 | 1 | 16 |
| **Fva1** | 0.125 | 0.25 | 1 | 16 |
| **Btr1** | 0.25 | 0.25 | 1 | 8 |
| **Pdi2*** | 0.25 | 0.25 | 2 | 8 |
| **Fva2** | 0.125 | 0.5 | 1 | 16 |
| **Dia1** | 0.25 | 2 | 2 | 64 |
| **Dno1** | 0.25 | 1 | 1 | 16 |
| **Dma1*** | 0.5 | 1 | 1 | 32 |
| **Vge1*** | 0.5 | 1 | 1 | 32 |
| **Mro1** | 0.5 | 2 | 1 | 32 |
| **Mge1** | 0.5 | 4 | 1 | ND |
| **Nme2** | 0.5 | 8 | 2 | ND |
| **Vma1*** | 0.5 | 4 | 2 | ND |
| **Fva3** | 1 | 8 | 1 | ND |
| **Ssp1*** | 1 | 16 | 2 | ND |
| **Btr2** | 1 | 8 | ND | ND |
| **Oab1** | 2 | 32 | ND | ND |
| **Obi1** | 2 | 16 | ND | ND |
| **Afl1** | 16 | >64 | ND | ND |
| **Bvo1** | >64 | >64 | ND | ND |
| **Baf1** | >64 | >64 | ND | ND |
| **Mqu1** | ND | ND | ND | ND |
| **Ado1** | ND | ND | ND | ND |
| **Nme1** | ND | ND | ND | ND |
| **Baf2** | ND | ND | ND | ND |
| **Baf3** | ND | ND | ND | ND |

MIC values are color-coded in a gradient: dark to light blue: 0.125–1 μM, yellow to brown: 2–8 μM, red: 16–64 μM, gray: >64 μM or not determined. The MIC experiments were performed in triplicate.
Peptides that have been described before are marked with asterisk.

the ribosome in a functional state corresponding to the post-peptide hydrolysis configuration of the PTC with RF1 and deacylated tRNA$^{Phe}$ in the A and P sites, respectively. The obtained Fourier maps revealed strong, continuous, and well-resolved electron density of the PrAMPs in the PTC-proximal part of the NPET, allowing us to build molecular models for the 14-residue-long C-terminal segments of Fva1 (Fig. 5A–C), Api (Fig. 5D,E), and Api137 (Fig. EV1B,C) peptides (residues 7–20 for Fva1 and residues 5–18 for Api and Api137). Residues 1–6 of Fva1 and residues 1–4 of Api and Api137 were invisible in the electron density maps, suggesting they are flexible and likely assume an ensemble of conformations in the NPET.

Superposition of the previously reported cryo-EM structure of Api137 complexed to the *E. coli* ribosome (Chan et al, 2020; Data

ref: Chan et al, 2020) with the new structure of the *T. thermophilus* ribosome in complex with the same PrAMP reveals no significant differences in the overall path of the peptide backbone as well as the placements of the visible side chains (Fig. EV2) validating the use of the *T. thermophilus* ribosome for structural studies of the PrAMPs and showing that Api-like PrAMPs bind to ribosomes from different bacteria in a similar way.

All three PrAMPs that we tackled in this work structurally (Fva1, Api, and Api137) are observed bound in the NPET in the extended conformations with their C-termini oriented toward the PTC and the N-termini located near the tunnel constriction, formed by the loops of ribosomal proteins uL4 and uL22 (Fig. 6A–C). Comparison of the obtained structures of

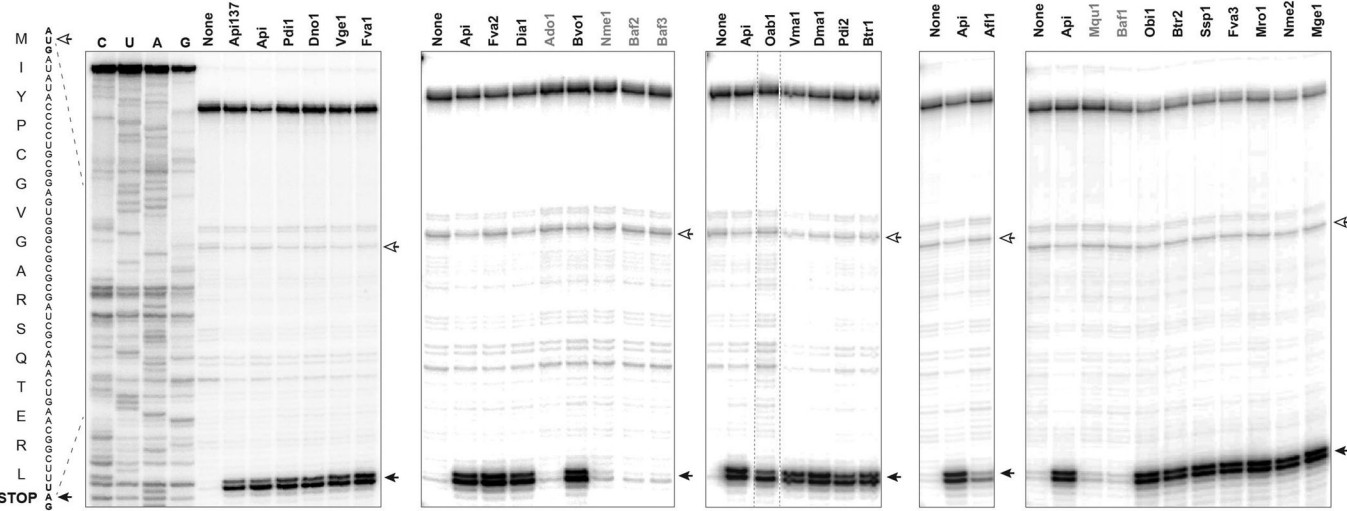

**Figure 3. In vitro toeprinting assays showing the ability of the tested PrAMPs to stall ribosomes at the stop codon of a model gene.**

The toeprint band corresponding to ribosomes arrested at the stop codon is indicated with a black arrow. The open arrow indicates the start codon. The names of the PrAMPs that significantly deviate from the Api sequence are in gray. Lanes marked as 'None' represent control samples where no PrAMP was added to the cell-free transcription–translation reaction. The sequence of the modified *E. coli yrbA* gene and the encoded protein sequence is shown on the left. Sequencing lanes are labeled as C, U, A, G. Dotted lines around the lane with the Oab1 peptide indicate the removed gel lanes that contained unrelated samples. Shown are representative gels from two independent experiments.

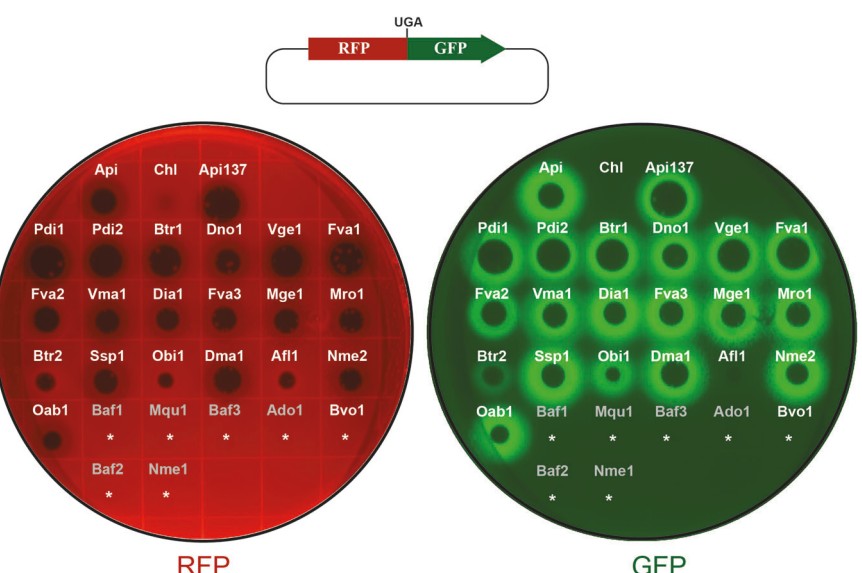

**Figure 4. Newly identified PrAMPs cause stop codon readthrough in bacterial cells.**

Top: In-frame fused genes encoding the reporter proteins RFP and GFP in the pRXG plasmid (Monk et al, 2017) are separated by an UGA stop codon. Bottom left: subinhibitory concentrations of the tested PrAMPs do not affect the production of RFP. Bottom right: green halo, indicative of GFP production, demonstrates the ability of some of the tested PrAMPs to induce, at subinhibitory concentrations, stop codon readthrough in bacterial cells. Api and the modified Api137 (Berthold et al, 2013) were spotted for comparison. The non-peptide ribosome-targeting antibiotic chloramphenicol (Chl), that does not induce stop codon readthrough, was included as a negative control. Shown are representative plates from three independent experiments.

ribosome-bound Fva1, Api, and Api137 reveals that their overall conformations and interactions within the NPET are very similar (Fig. 6C), except for a slight deviation in the paths of the four most distal visible N-terminal amino acid residues (Fig. 6C). Since most of the sequence differences between Api and Fva1 peptides reside in

their N-termini (Fig. 1D), which were not resolved in our structures, we conclude that the N-terminal elements, while being important for antibacterial activity (Baliga et al, 2021; Huang et al, 2024), do not affect the placement of the C-terminal "warhead" of Api-like PrAMPs in the ribosomal tunnel. Moreover, Api137 differs

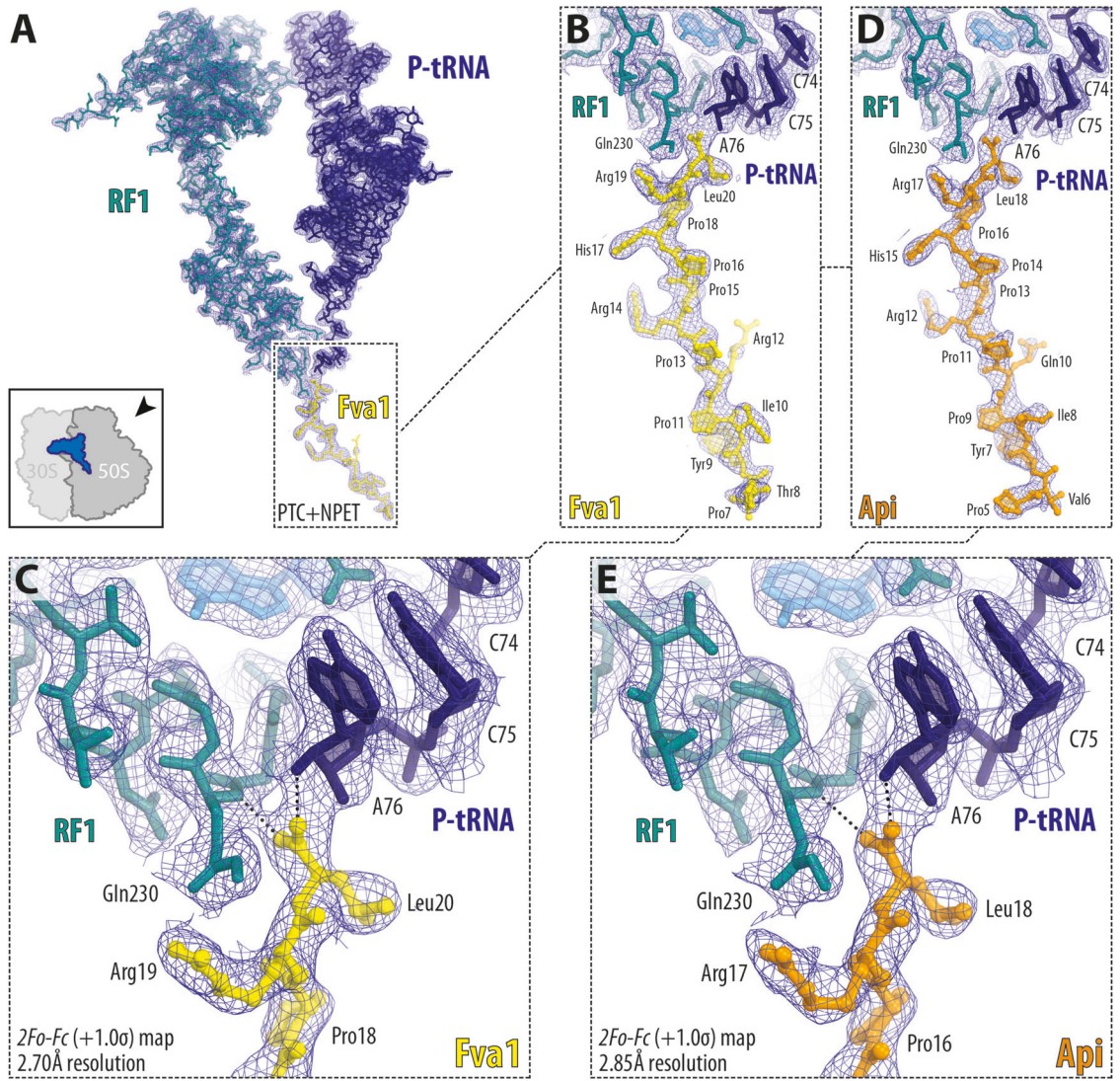

**Figure 5. Electron density maps of ribosome-bound Fva1 and Api peptides.**

(A–E) $2F_o$-$F_c$ Fourier electron density map (blue mesh) of Fva1 (A–C) or Api (D, E) in complex with the *T. thermophilus* ribosome. The refined models of Fva1 (yellow) or Api (orange) are displayed in their respective electron densities after the refinement contoured at 1.0σ. The adjacent ribosome-bound release factor 1 (RF1) and deacylated P-site tRNA (P-tRNA) are shown in teal and navy blue, respectively.

from Api by carrying N,N,N',N'-tetramethylguanidino ornithine instead of the N-terminal glycine of the WT Api and by having Gln10Arg substitution. Neither Gln10 of Api (Fig. 5D) nor Arg10 of Api137 (Fig. EV1) is well-resolved in the obtained electron density maps, suggesting that these residues are flexible and are unlikely to interact strongly with the ribosome or contribute to the on-target activity of these peptides.

The C-terminal residues of Fva1, Api, or Api137 form an extensive network of hydrogen bonds (H-bonds) and π-π stacking interactions with elements of the ribosome and its ligands (Fig. 6D–F), rationalizing the evolutionary conservation of the C-terminal sequences among the Api-like Type II PrAMPs. Specifically, one of the oxygens of the carboxylic group of the C-terminal Leu residue is within H-bond distance from the 2'- and

3'-hydroxyls of the A76 residue of the deacylated tRNA in the P site, whereas the other oxygen forms H-bond with the main-chain NH group of the Gln230 residue in the conserved GGQ motif of RF1 (Figs. 6D and EV1C). Moreover, the side chain of the functionally critical penultimate Arg residue of Fva1/Api/Api137 forms an H-bond with the side chain of the same Gln230 residue of RF1 in the A site (Fig. 6D). Thus, target engagement for Type II PrAMPs is driven not only by interactions with elements of the ribosome itself but also with ribosomal ligands, such as tRNAs and RFs, emphasizing the critical importance of using functionally relevant complexes for structural studies of ribosome-bound inhibitors of translation.

Perhaps the most peculiar interactions of Fva1, Api, and Api137 peptides are formed with nucleotides G2061, and A2062 of the 23S

**Table 2. X-ray data collection and refinement statistics.**

| Crystal(s) | *Tth* 70S ribosome in complex with mRNA, A-site RF1, P-site tRNA^Phe, and Fva1 PDB entry 9XXX | *Tth* 70S ribosome in complex with mRNA, A-site RF1, P-site tRNA^Phe, and Api1B PDB entry 9YYY | *Tth* 70S ribosome in complex with mRNA, A-site RF1, P-site tRNA^Phe, and Api137 PDB entry 9ZZZ |
|---|---|---|---|
| **Diffraction data** | | | |
| Synchrotron | BNL-NSLSII | BNL-NSLSII | ANL-APS |
| Beamline | 17ID-1 (AMX) | 17ID-2 (FMX) | 24ID-C |
| Space group | $P2_12_12_1$ | $P2_12_12_1$ | $P2_12_12_1$ |
| Unit cell dimensions, Å ($a \times b \times c$) | 210.39 × 451.41 × 623.94 | 210.35 × 451.32 × 625.61 | 209.6 × 450.81 × 624.08 |
| Wavelength, Å | 1.033202 | 0.979338 | 0.97911 |
| Resolution range (outer shell), Å | 366-2.70 (2.77-2.70) | 313-2.85 (2.92-2.85) | 312-2.70 (2.77-2.70) |
| I/σI (outer shell) | 7.05 (0.87) | 6.42 (1.02) | 8.51 (0.89) |
| Resolution at which I/σI = 2, Å | 2.95 | 3.05 | 2.95 |
| Completeness (outer shell), % | 96.9 (97.3) | 99.1 (98.9) | 99.1 (98.8) |
| $R_{merge}$ (outer shell)% | 23.6 (177.2) | 29.5 (174.3) | 20.5 (202.4) |
| No. of crystals used | 1 | | 1 |
| No. of reflections used: Total | 8,376,402 | 8,078,242 | 9,461,928 |
| Unique | 1,553,365 | 1,355,591 | 1,580,937 |
| Redundancy (outer shell) | 5.39 (5.27) | 5.96 (6.08) | 5.99 (5.54) |
| **Refinement** | | | |
| Resolution range of used data, Å | 37-2.70 | 102-2.85 | 153-2.70 |
| No. of reflections used | 1,550,230 | 1,355,324 | 1,580,648 |
| $R_{work}/R_{free}$, % | 22.1/27.6 | 21.3/27.0 | 221/28.0 |
| *No. of Non-Hydrogen Atoms* | | | |
| Protein | 95,231 | 95,230 | 95,231 |
| Ligand (RNA/Ions) | 200,706 | 200,709 | 200,709 |
| Waters | 3094 | 3098 | 3098 |
| *B factors* | | | |
| Protein | 53.8 | 56.4 | 60.3 |
| Ligand (RNA/Ions) | 50.4 | 52.9 | 57.3 |
| Waters | 34.9 | 37.2 | 41.0 |
| *Deviations from ideal values (RMSD)* | | | |
| Bond, Å | 0.009 | 0.009 | 0.010 |
| Angle, degrees | 1.417 | 1.403 | 1.566 |

rRNA (Fig. 6E,F). The same exact network of H-bonds has been identified in our recent studies reporting the structures of 70S ribosome in the pre-transpeptidation state containing various peptidyl-tRNAs in the P site (Aleksandrova et al, 2024; Syroegin et al, 2022a, 2023). These H-bonds were suggested to be critically important for tightly coordinating growing polypeptides in the NPET (Syroegin et al, 2023). In the case of type II PrAMPs, the same H-bonds likely provide additional stabilization of the peptides in the NPET. Overall, our structural analysis of the functional ribosome-PrAMP complexes demonstrated that even significant variation in the length and sequence of the N-terminal segment of the Api-like PrAMPs does not disrupt critical interactions of the functionally important C-terminal segment of the peptide with the ribosomal PTC, NPET, tRNA, and RF. It is also evident that PrAMPs establish very similar interactions with the ribosomes of phylogenetically distant Gram-negative bacterial species, such as *E. coli* and *T. thermophilus*.

## Discussion

In this work, we identified 71 PrAMPs, most of them resembling Api, encoded as polyproteins in insect genomes. Overall, the diversity of Api homologs reflects the adaptability and versatility of these antimicrobial peptides in different organisms and their role in innate immunity.

The diversity of the Api-like Type II PrAMPs is likely driven by the spectrum of bacterial species infecting the corresponding hosts. In this regard, it is noteworthy that while ribosome-targeting PrAMPs are encoded in genomes of various insects and mammals, Api-like peptides are found only in wasps and bees. The reasons for such narrow species confinement are unclear but could be related to the spectrum of bacterial pathogens infecting specifically these hosts. Previous studies have shown that minor variations in the sequence differentially affect the activity of PrAMPs against divergent bacteria (Hanson et al, 2023; Lazzaro et al, 2020;

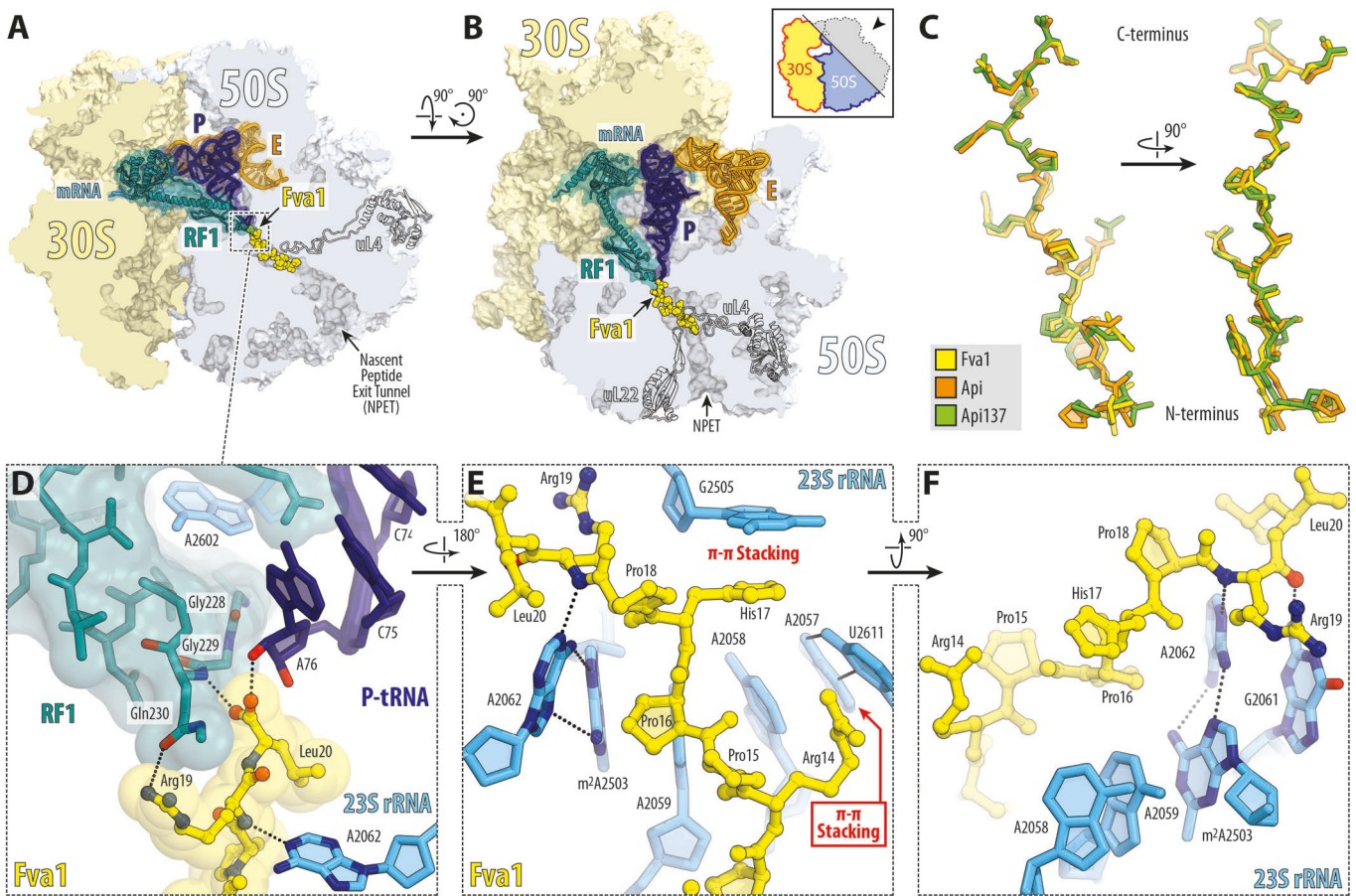

**Figure 6. Structure of Fva1 in complex with the wild-type _T. thermophilus_ 70S ribosome.**

(**A**, **B**) Overview of the PrAMP-binding site in the ribosome, viewed as a cross-cut section through the nascent peptide exit tunnel (NPET). The 30S subunit is shown in light yellow, the 50S subunit is shown in light blue. The mRNA is in blue, and the A-site release factor 1 (RF1), P-, and E-site tRNAs are colored teal, navy blue, and orange, respectively. Fva1 PrAMP is yellow. (**C**) Superposition of the structures of ribosome-bound Fva1 (yellow), Api (orange), and Api137 (green). (**D**) Close-up view of Fva1 C-terminus, highlighting the interactions of this Type II PrAMP with RF1, P-site tRNA, and 23S rRNA (_E. coli_ numbering of the rRNA nucleotides is used throughout). (**E**, **F**) Interactions of C-terminal residues of Fva1 with the 23S rRNA nucleotides. Note that by forming a characteristic H-bond with the main-chain amino group of the penultimate Arg19 residue of Fva1, nucleotide A2062 of the 23S rRNA coordinates it the same way as it would in a nascent peptide.

Perlmutter et al, 2024). Thus, it appears that the activity of PrAMPs can be easily evolutionarily adjusted by only a few mutations to protect the host against newly emerging pathogens.

All the tested active peptides could arrest the terminating ribosome and induce stop codon readthrough exhibiting the mode of action characteristic of Type II PrAMPs. While these PrAMPs exhibited significant variations in the structures of their N-terminal segments, their C-terminal sequences are highly conserved. Therefore, it appears that the C-terminal domain defines the mode of action of these PrAMPs. This conclusion is vividly illustrated by our high-resolution structures of Fva1 and Api bound to the bacterial ribosome. Irrespective of the N-terminal sequence difference between these two PrAMPs, their C-terminal domains establish the same interactions with the ribosomal NPET, deacylated P-site tRNA, and A-site RF1. Extrapolating these findings, we expect that the placement of the C-terminal segment of all the active Api-like PrAMPs in the ribosome is likely uniform despite variations in the peptides' N-terminal sequences revealing the C-terminal domain as the main pharmacophore of these Type II PrAMPs. Building upon these observations, we favor the view that the improved activity of Fva1

relative to that of Api results from more favorable interactions with the SbmA transporter primarily responsible for delivering Api-like PrAMPs into the bacterial cytoplasm (Mattiuzzo et al, 2007), rather than from increased affinity of the PrAMP to the ribosome. The importance of the C-terminal segments of the Api-like PrAMPs for their activity and its uniform placement in the ribosome are consistent with the results of the screening of the in vivo expressed single-mutant Api gene libraries, where the majority of the mutations within the C-terminal sequence diminished the inhibitory potency of the PrAMPs (Baliga et al, 2021; Huang et al, 2024). Nevertheless, analyzing the activity of new natural Api-like PrAMPs we found that some C-terminal mutations could be tolerated. Thus, PrAMPs Oab1, Obi1, Afl1, and Bvo1 retain the antibacterial and/or ribosome-targeting activities despite having one or even two substitutions in the conserved C-terminal domain. Thus, hypothetically, it could be possible to modify the Type II PrAMP pharmacophore to further improve the on-target activity of these antibiotics.

The placement of the ribosome-bound Fva1/Api peptides in the NPET perfectly matches that of the growing nascent peptide chain

attached to the P-site tRNAs. In other words, although Fva1/Api/Api137 peptides were added to the complex in trans, they are positioned in the NPET as if they were synthesized by the very same ribosome *in cis*, hydrolyzed off the tRNA carrier in the presence of RF1 but, instead of immediately leaving the tunnel as most other polypeptides would do, become trapped in the NPET. This observation explains why, in addition to acting in trans, Type II PrAMPs can likely also act *in cis* and trap RFs on the ribosome during the endogenous expression of their genes in bacteria (Baliga et al, 2021; Mangano et al, 2023; Taguchi et al, 2009; Taguchi et al, 1994; Taguchi et al, 1996).

Previous studies showed that Api-mediated trapping of RFs on the post-release ribosome results in depletion of the pool of the available RFs in the cell that eventually leads to greatly enhanced stop codon readthrough (Florin et al, 2017; Mangano et al, 2020). Similar to Api, the majority of the newly identified and tested active PrAMPs readily stimulated bypass of the stop codon in the *rfp-gfp* reporter (Fig. 4). Noteworthy, the extent of stop codon bypass varies between different PrAMPs and, at the qualitative level, there appears to be no direct correlation between the ability of the PrAMP to inhibit cell growth and the extent of stop codon readthrough (compare, for example, the effects of Btr2 and Ssp1 or Obi1 and Afl1 peptides in Fig. 4). Hence, the intracellular action of Type II PrAMPs is likely more nuanced and involves additional effects that await exploration (Lauer et al, 2024).

Importantly, the antimicrobial efficacy of several of the tested Api-like PrAMPs exceeds that of wt Api by 4- to 16-fold. Comparing the most active PrAMPs with those that retain the conserved C-terminal segment of Api but exhibit lower antibacterial activity shows several features in their N-terminal regions that seem to correlate with the improved antibacterial properties. Specifically, most of the more active peptides, including Pdi1, Pdi2, Fva1, Fva2, Btr1, Dno1, and Dia1 (Fig. 1D), feature a one- or two amino acids-long N-terminal extension compared to Api. In addition, the net charge of seven of the more active peptides ranged from +4 to +6, with the average charge being +4.7, whereas the less active peptides had a net charge in the +3 to +4 range, with the average charge being +3.4 (Table EV1). Thus, the longer PrAMPs with a higher positive charge appear to be more potent antibacterials. Because variations in the length and net charge come primarily from alterations in the properties of the PrAMPs' N-termini that are dispensable for binding to the ribosome, it is tempting to think that these activity-related trends reflect the SbmA-mediated intracellular uptake of the PrAMPs. The detailed correlation of the in vivo antibacterial and in vitro on-target activities of various PrAMPs could illuminate the currently unknown specificity of this enigmatic peptide transporter and can be exploited in the future for a knowledge-based improvement of the antibiotic activity of Type II PrAMPs.

Our findings emphasize that the fast-growing genomic data represent a vast reservoir of yet unexplored antimicrobial peptides. Being the first known specific inhibitors of translation termination, Type II PrAMPs hold significant promise as a starting point for developing new clinically useful antibacterials. The diverse sequences and nuanced properties of these inhibitors underscore their potential as valuable antimicrobial agents and warrant further exploration. The identification of new Type II PrAMPs with the activity exceeding any other previously known inhibitors of this class is an important step toward the development of Type II PrAMPs into clinically valuable antibiotics.

# Methods

## Reagents and tools table

| Reagent/resource | Reference or source | Identifier or catalog number |
| --- | --- | --- |
| **Experimental models** | | |
| *E. coli*, strain BL21 | New England Biolabs | C2530H |
| Plasmid pRXG(UGA) | Monk et al, 2017 | N/A |
| *K. pneumoniae*, strain AR-0112 | CDC & FDA Antimicrobial Resistance Isolate Bank | SAMN04014953 |
| **Oligonucleotides and other sequence-based reagents** | | |
| DNA oligonucleotides | Integrated DNA Technologies | Custom order |
| **Chemicals, enzymes, and other reagents** | | |
| Lysogeny Broth (LB) | BD DIFCO™ | SKU: 240210 |
| Tryptone Soy Broth (TSB) | BD DIFCO™ | SKU: 211825 |
| Mueller Hinton Broth | BD DIFCO™ | SKU: 211443 |
| M9 Salts | Sigma-Aldrich | M6030 |
| Growth medium RPMI 1640 | Thermo Fisher Scientific | A4192301 |
| Chloramphenicol | Fisher Scientific | BP904-100 |
| Fetal Bovine Serum | Thermo Fisher Scientific | A3160501 |
| Q5® Hot Start High-Fidelity DNA Polymerase | New England Biolabs | M0493 |
| PURExpress® In Vitro Protein Synthesis Kit | New England Biolabs | E6800L |
| Peptides | NovoPro | Custom order |
| β-Mercaptoethanol | BioRad | 1610710 |
| HEPES | Sigma-Aldrich | H4034 |
| Potassium chloride | Sigma-Aldrich | P9541 |
| 1 M Magnesium acetate solution | Sigma-Aldrich | 63052 |
| Water | Sigma-Aldrich | W4502 |
| L-Arginine monohydrochloride | Sigma-Aldrich | A5131 |
| 1 M Trizma hydrochloride, pH 7.6 | Sigma-Aldrich | T2444 |
| Polyethylene Glycol 20,000 | Hampton Research | HR2-609 |
| 2-Methyl-2,4-pentanediol | Hampton Research | HR2-627 |
| **Software** | | |
| Image Lab software | BioRad | 12012931 |
| IQTree | http://iqtree.cibiv.univie.ac.at/ | |
| iTOL | https://itol.embl.de/ | |
| Geneious Prime | Geneious | version 2023.2.1 |
| ChemDraw | Revvity Signals | https://revvitysignals.com/products/research/chemdraw |

| Reagent/resource | Reference or source | Identifier or catalog number |
|---|---|---|
| XDS software package (version from Jan 10, 2022) | Kabsch, 2010 | xds.mr.mpg.de |
| CCP4 program suite (version 7.0) | McCoy et al, 2007 | www.ccp4.ac.uk/html/phaser.html |
| PHENIX software (version 1.17) | Adams et al, 2010 | https://phenix-online.org/ |
| COOT (version 0.8.2) | Emsley and Cowtan, 2004 | www.2.mrc-lmb.cam.ac.uk/personal/pemsley/coot/ |
| PRODRG online software | Schuttelkopf and van Aalten, 2004 | https://www.ccp4.ac.uk/html/cprodrg.html |
| PyMOL Molecular Graphics System software (version 1.8.6) | Schrödinger | www.pymol.org |
| **Other** | | |
| Microplate reader Infinite M200Pro | Tecan | # 30050303 |
| BioRad ChemiDoc MP | BioRad | # 12003154 |

## Identifying Api homologs by mining nucleotide and protein databases

The protein sequence GNNRPVYIPQPRPPHPRL of Apidaecin 1b (Api) from *Apis mellifera* (Casteels et al, 1989) was used as a query with the BLASTP algorithm of the NCBI BLAST suite, to search the non-redundant protein sequences database. Remote homologs were identified with the Api sequence, as well as the complete Api precursor protein (GenBank accession number CAA51168.1) as a query, by running PSI-BLAST for two iterations. Independently, the *A. mellifera* Apid14 mRNA (GenBank accession number X72575.1) was used as a query in BLASTN searches, and results were translated into the amino acid sequences of the hits. Most of the initial hits corresponded to the predicted pre-pro-proteins encoding multiple PrAMP repeats. The boundaries of the predicted Api-like peptides and other co-encoded PrAMPs were deduced on the basis of the previously identified protease processing signals in the Api-encoding pre-pro-protein from the *A. mellifera* genome (Casteels-Josson et al, 1993). The PrAMP sequences were extracted for further analysis.

The aforementioned approach yielded 71 probable non-redundant PrAMPs across multiple species of bees and wasps (Table EV1). The multiple sequence alignments of PrAMPs were conducted using Geneious Prime software (version 2023.2.1) with the MUSCLE algorithm (Edgar, 2004). The alignments were additionally manually adjusted. For phylogenetic analysis, the aligned sequences were further trimmed to eliminate ambiguous regions before IQTree building (Minh et al, 2020) (http://iqtree.cibiv.univie.ac.at/) based on the maximum likelihood method. Type I PrAMPs Metalnikowin, Oncocin, and Pyrrhocoricin were also included in tree generation, the proline-arginine rich cathelicidin PR-39 from pig were included as an outgroup. iTOL

(Letunic and Bork, 2024) (https://itol.embl.de/) was used for tree visualization and editing.

Twenty-six peptides containing 20 amino acids or less, chosen on the bases of their divergence from the Api sequence and from each other, were chemically synthesized (NovoPro Biosciences, China) at >85% purity. These peptides were then used in microbiological and biochemical assays at concentrations adjusted for the content of the target peptide in the preparation.

## Drop-diffusion test for antimicrobial activity

Bacterial cells, *Escherichia coli* strain BL21 and *Klebsiella pneumoniae* strain AR-0112, were grown overnight in lysogeny broth (LB) medium at 37 °C with shaking. The overnight cultures were diluted 100-fold into fresh LB medium and grown at 37 °C with shaking until culture density reached $A_{600} \sim 0.8$. Culture (5 mL) was poured onto agar plates, prepared with threefold diluted Tryptone Soy Broth (33% TSB) media, and evenly spread. Excess liquid was aspirated, and plates were allowed to dry. Either 2 µl of 2 mM peptide solution or 1 µl of 1 mg/mL chloramphenicol solution were applied to the plate surface. The plates were incubated at 37 °C for 16 h and photographed.

## Determination of the PrAMPs' minimal inhibitory concentrations

The minimal inhibitory concentration (MIC) test for the *E. coli* laboratory strain BL21 and the *K. pneumoniae* multi-drug-resistant clinical strain AR-0112 (CDC and FDA Antibiotic Resistance Isolate Bank) was performed using the broth microdilution method, adapted from a previously described procedure (Wiegand et al, 2008). The MIC tests were conducted either in 33% Tryptic Soy Broth (TSB) (Berthold et al, 2013), or in RPMI/Serum (prepared by combining Roswell Park Memorial Institute medium 1640, Mueller Hinton Broth, and fetal bovine serum in the proportion 100:5:20, v/v/v) (Belanger et al, 2020).

## Stop codon readthrough assay

Stop codon readthrough activity of PrAMPs was determined as previously described (Mangano et al, 2023). Briefly, *E. coli* strain BL21 cells transformed with the pRXG(UGA) plasmid (Monk et al, 2017) were plated as described above for the drop-diffusion test, except that the bacterial lawn was formed on M9 medium/agar plates supplemented with 50 µg/ml of Kan and 0.2% glucose. Peptide solutions (2 µL of 2 mM stocks) were applied. The plates were incubated at 37 °C for 16 h and imaged in Cy2 (GFP) or Cy3 (RFP) channels using the BioRad ChemiDoc Touch imaging system. The images were processed using Image Lab software (BioRad).

## Toeprinting analysis

Toeprinting experiments were conducted following a previously established procedure (Florin et al, 2017; Orelle et al, 2013). Briefly, the DNA templates containing the modified and truncated *E. coli yrbA* gene ending with the TAG stop codon were expressed in the PURExpress cell-free transcription–translation system (Shimizu et al, 2001) (New England Biolabs). When needed, the 5 µL reactions were

supplemented with 50 μM of the PrAMP solutions. The location of the stalled ribosome was determined by reverse transcriptase-catalyzed extension of the NV1 primer (GGTTATAATGAATTTTGCTTAT-TAAC). The reaction products were resolved in 6% sequencing gels alongside with the sequence reactions.

## X-ray crystallographic structure determination

Wild-type 70S ribosomes from *T. thermophilus* (strain HB8) were prepared as described previously (Polikanov et al, 2015; Polikanov et al, 2014; Syroegin et al, 2022b). Synthetic mRNA with the sequence 5′-GGCAA*GGAGG*UAAAA<u>UUCUAA</u>UAA-3′ containing Shine-Dalgarno sequence (italicized) followed by the P-site phenylalanine (UUC) and A-site stop (UAA) codons (underlined) was obtained from Integrated DNA Technologies (Coralville, IA, USA). Ribosome complexes with mRNA and tRNA were formed by mixing 5 μM *T. thermophilus* 70S ribosomes with 10 μM mRNA and incubation at 55 °C for 10 min, followed by the addition of 50 μM *T. thermophilus* RF1 and 20 μM deacylated tRNA^Phe for A and P sites, respectively. For co-crystallization of the ribosome with Api137, Api, or Fva1, peptides were added to a final concentrations of 100 μM, and the complex was left at room temperature for an additional 15 min prior to crystallization. Ribosome complexes were prepared in a buffer containing 5 mM HEPES-KOH (pH 7.6), 50 mM KCl, 10 mM $NH_4Cl$, and 10 mM $Mg(CH_3COO)_2$, and then crystallized in a buffer containing 100 mM Tris-HCl (pH 7.6), 2.9% (w/v) PEG-20K, 9–10% (v/v) MPD, 175 mM arginine, 0.5 mM β-mercaptoethanol. Crystals were grown by the vapor diffusion method in sitting drops at 19 °C, stabilized and cryo-protected stepwise using a series of buffers with increasing MPD concentrations (25%, 30%, 35%) until reaching the final concentration of 40% (v/v) as described previously (Polikanov et al, 2015; Svetlov et al, 2019; Svetlov et al, 2021; Syroegin et al, 2022a, 2023; Syroegin et al, 2022b; Tereshchenkov et al, 2018).

Collection and processing of the X-ray diffraction data, model building, and structure refinement were performed as described in our previous reports (Polikanov et al, 2015; Svetlov et al, 2019; Svetlov et al, 2021; Syroegin et al, 2022a, 2023; Syroegin et al, 2022b; Tereshchenkov et al, 2018). Diffraction data were collected at beamlines 24ID-C and 24ID-E at the Advanced Photon Source (Argonne National Laboratory) and beamlines 17ID-1 and 17ID-2 of the National Synchrotron Light Source II (Brookhaven National Laboratory). A complete dataset was collected using 0.979-Å (APS) or 1.033-Å (NSNL-II) X-ray irradiation at 100 K from multiple regions of the same crystal, using 0.3-degree oscillations. Raw data were integrated and scaled using the XDS software package (version from Jan 10, 2022). Molecular replacement was performed using PHASER from the CCP4 program suite (version 7.0) (Kabsch, 2010). The initial search model was generated by replacing the A-site Phe-tRNA^Phe in previously published structures of the *T. thermophilus* 70S ribosome with bound mRNA and aminoacylated tRNAs with the PDB entry 6XHW (Svetlov et al, 2021; Data ref: Svetlov et al, 2021) and with the ribosome-bound RF1 taken from PDB entry 4V63 (Laurberg et al, 2008; Data ref: Laurberg et al, 2008). Initial molecular replacement solutions were refined by rigid-body refinement with the ribosome split into multiple domains, followed by positional and individual B-factor refinement using the PHENIX software (version 1.17) (Adams et al, 2010). Non-crystallographic symmetry restraints were applied to four

parts of the 30S ribosomal subunit (head, body, spur, and helix 44) and four parts of the 50S subunit (body, L1-stalk, L10-stalk, and C-terminus of the L9 protein). The initial PDB model of Api137 peptide bound to *E. coli* 70S ribosome was taken from the PDB entry 6YSS (Chan et al, 2020; Data ref: Chan et al, 2020). The final PDB models of ribosome-bound Fva1, Api, and Api137 peptides were built in Coot (version 0.8.2) (Emsley and Cowtan, 2004). All figures showing atomic models were rendered using PyMOL Molecular Graphics System software (version 1.8.6, Schrödinger, www.pymol.org).

## Data availability

Coordinates and structure factors of the *T. thermophilus* ribosome in complex with mRNA, decylated P-site tRNA^Phe, A-site *T. thermophilus* RF1, and PrAMPs, were deposited in the RCSB Protein Data Bank with the following accession codes: 9D7R for Fva1, 9D7S for Api1b, and 9D7T for Api137.

The source data of this paper are collected in the following database record: biostudies:S-SCDT-10_1038-S44319-024-00277-5.

## Peer review information

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

## Acknowledgements

The authors thank the staff at NE-CAT beamlines 24ID-C and 24ID-E for help with data collection, especially Drs. Malcolm Capel, Frank Murphy, Surajit Banerjee, Igor Kourinov, David Neau, Jonathan Schuermann, Narayanasami Sukumar, Anthony Lynch, James Withrow, Kay Perry, Ali Kaya, and Cyndi Salbego. This work is based upon research conducted at the Northeastern Collaborative Access Team beamlines, which are funded by the National Institute of General Medical Sciences from the National Institutes of Health [P30-GM124165 to NE-CAT]. The Eiger 16 M detector on 24-ID-E beamline is funded by an NIH-ORIP HEI grant [S10-OD021527 to NE-CAT]. This research used resources of the Advanced Photon Source, a US Department of Energy (DOE) Office of Science User Facility operated for the DOE Office of Science by Argonne National Laboratory under Contract No. DE-AC02-06CH11357. This work is also based upon research conducted at the Center for BioMolecular Structure beamlines (17ID-1 and 17ID-2), which are primarily supported by the National Institute of General Medical Sciences from the National Institutes of Health [P30-GM133893], and by the DOE Office of Biological and Environmental Research [KP1605010]. NSLS2 is a U.S. DOE Office of Science User Facility operated under Contract No. DE-SC0012704. This publication resulted from the data collected using the beamtime obtained through NE-CAT BAG proposal #311950. This work was supported by the National Institute of Allergy and Infectious Diseases of the National Institutes of Health [R01-AI162961 to ASM, NV-L, and YSP], National Institute of General Medical Sciences of the National Institutes of Health [R01-GM132302 and R35-GM151957 to YSP and R35-GM127134 to ASM], the National Science Foundation [MCB-1907273 to YSP], the Illinois State startup funds [to YSP], and the Swedish Research Council (Vetenskapsrådet) [2019-01085, 2022-01603 and 2023-02353 [to GCA].

## Author contributions

**Weiping Huang**: Formal analysis; Investigation; Visualization; Methodology; Writing—original draft; Writing—review and editing. **Chetana Baliga**: Conceptualization; Formal analysis; Investigation; Writing—original draft; Writing—review and editing. **Elena V Aleksandrova**: Investigation; Visualization; Methodology. **Gemma Atkinson**: Formal analysis; Visualization; Writing—review and editing. **Yury S Polikanov**: Conceptualization; Data curation; Formal analysis; Supervision; Funding acquisition; Validation; Visualization; Writing—original draft; Writing—review and editing. **Nora Vázquez-Laslop**: Conceptualization; Supervision; Funding acquisition; Validation; Writing—original draft; Writing—review and editing. **Alexander S Mankin**: Conceptualization; Supervision; Funding acquisition; Validation; Visualization; Writing—original draft; Writing—review and editing.

Source data underlying figure panels in this paper may have individual authorship assigned. Where available, figure panel/source data authorship is listed in the following database record: biostudies:S-SCDT-10_1038-S44319-024-00277-5.

## Disclosure and competing interests statement

The authors declare no competing interests.

# Expanded View Figures

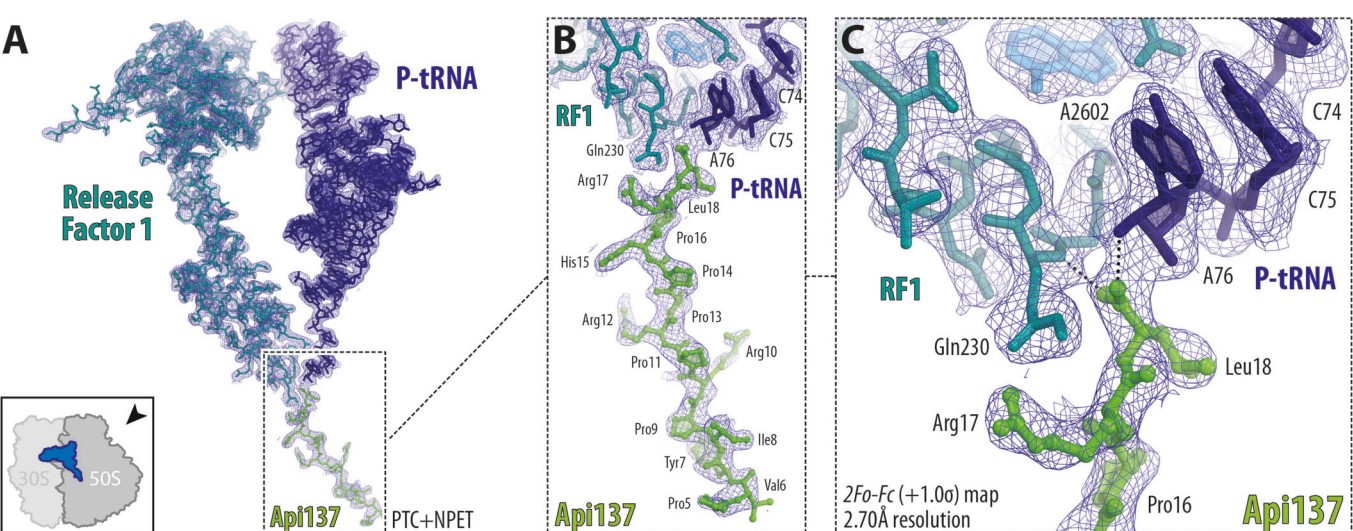

**Figure EV1.  Electron density map of the ribosome-bound Api137 peptide.**

(A–C) $2F_o$-$F_c$ Fourier electron density map of Api137 in complex with the *T. thermophilus* 70S ribosome (blue mesh). The refined model of Api137 (green) is displayed in its respective electron density after the refinement contoured at 1.0σ. The adjacent ribosome-bound release factor 1 (RF1) and deacylated P-site tRNA (P-tRNA) are shown in teal and navy blue, respectively.

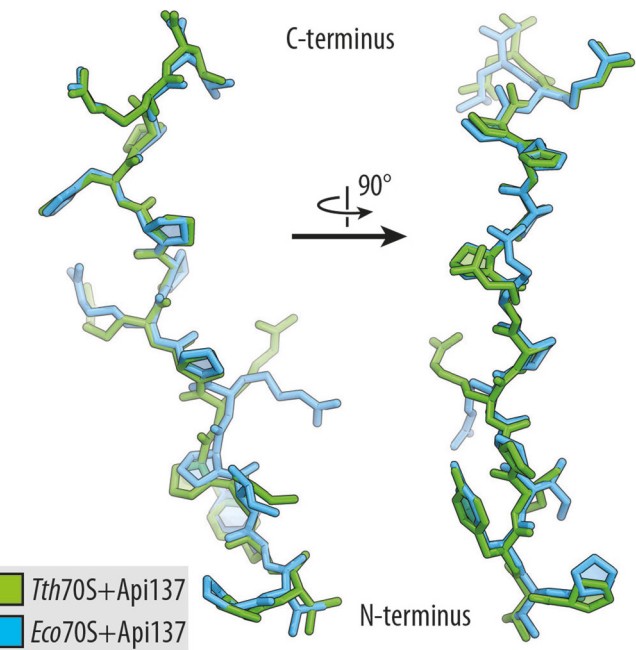

**Figure EV2. Comparison of Api137 bound to *T. thermophilus* and *E. coli* 70S ribosomes.**

Superposition of the new X-ray structure of Api137 in complex with the wild-type *T. thermophilus* ribosome (green) with the previous cryo-EM structure of Api137 in complex with ribosome from *E. coli* [blue, PDB entry 6YSS (Chan et al, 2020; Data ref: Chan et al, 2020)].

