## [Peer Review File · EMBO Reports]

Activity, structure, and diversity of Type II proline-rich antimicrobial peptides from insects

Weiping Huang, Chetana Baliga, Elena Aleksandrova, Gemma Atkinson, Yury Polikanov, Nora Vazquez-Laslop, and Alexander Mankin

Corresponding author(s): Alexander Mankin (shura@uic.edu), Nora Vazquez-Laslop (nvazquez@uic.edu), Yury Polikanov (yuryp@uic.edu)

Review Timeline:

Submission Date:	4th Jul 24
Editorial Decision:	6th Aug 24
Revision Received:	19th Aug 24
Editorial Decision:	3rd Sep 24
Revision Received:	15th Sep 24
Accepted:	19th Sep 24

Editor: Achim Breiling / Martina Rembold

Transaction Report:

Dear Dr. Mankin,

Thank you for the submission of your manuscript to EMBO reports. I have now received the reports from the three referees that were asked to evaluate your study, which can be found at the end of this email.

As you will see, the referees find the study interesting. However, referees #2 and #3 have several comments, concerns, and suggestions, indicating that a major revision of the manuscript is necessary to allow publication of the study in EMBO reports. As the reports are below, and all the concerns need to be addressed, I will not detail them further here.

Acceptance of your manuscript will depend on a positive outcome of a second round of review. It is EMBO reports policy to allow a single round of revision only and acceptance of the manuscript will therefore depend on the completeness of your responses included in the next, final version of the manuscript.

1) a .docx formatted version of the final manuscript text (including legends for main figures, EV figures and tables), but without the figures included. Figure legends should be compiled at the end of the manuscript text.

2) individual production quality figure files as .eps, .tif, .jpg (one file per figure), of main figures and EV figures. Please upload these as separate, individual files upon re-submission.

4) a complete author checklist, which you can download from our author guidelines

(<https://www.embopress.org/page/journal/14693178/authorguide>). Please insert page numbers in the checklist to indicate where the requested information can be found in the manuscript. The completed author checklist will also be part of the RPF.

5) that primary datasets produced in this study (e.g. RNA-seq, ChIP-seq, structural and array data) are deposited in an appropriate public database. If no primary datasets have been deposited, please also state this in a dedicated section (e.g. 'No primary datasets have been generated and deposited'), see below.

The accession numbers and database should be listed in a formal "Data Availability" section (placed after Materials & Methods) that follows the model below. This is now mandatory (like the COI statement). Please note that the Data Availability Section is restricted to new primary data that are part of this study. This section is mandatory. As indicated above, if no primary datasets have been deposited, please state this in this section

Data availability

8) Regarding data quantification and statistics, please make sure that the number "n" for how many independent experiments were performed, their nature (biological versus technical replicates), the bars and error bars (e.g. SEM, SD) and the test used to calculate p-values is indicated in the respective figure legends (also for EV figures and all those in an Appendix). Please also check that all the p-values are explained in the legend, and that these fit to those shown in the figure. Please provide statistical testing where applicable. Please avoid the phrase 'independent experiment', but clearly state if these were biological or technical replicates. Please also indicate (e.g. with n.s.) if testing was performed, but the differences are not significant. In case n=2, please show the data as separate datapoints without error bars and statistics. See also: <http://www.embopress.org/page/journal/14693178/authorguide#statisticalanalysis>

9) Please add scale bars of similar style and thickness to microscopic images, using clearly visible black or white bars (depending on the background). Please place these in the lower right corner of the images themselves. Please do not write on or near the bars in the image but define the size in the respective figure legend.

10) Please also note our reference format:

12) We now use CRediT to specify the contributions of each author in the journal submission system. CRediT replaces the author contribution section. Please use the free text box to provide more detailed descriptions and do NOT provide your final manuscript text file with an author contributions section. See also our guide to authors: <https://www.embopress.org/page/journal/14693178/authorguide#authorshipguidelines>

13) All Materials and Methods need to be described in the main text using our 'Structured Methods' format, which is required for all research articles. According to this format, the Materials and Methods section should include a Reagents and Tools Table (listing key reagents, experimental models, software, and relevant equipment and including their sources and relevant

identifiers), uploaded as separate file, followed by a Methods and Protocols section in which we encourage the authors to describe their methods using a step-by-step protocol format with bullet points, to facilitate the adoption of the methodologies across labs. More information on how to adhere to this format as well as downloadable templates (.doc) for the Reagents and Tools Table can be found in our author guidelines (section 'Structured Methods'):

14) Please order the manuscript sections like this, using these names:

Title page - Abstract - Keywords - Introduction - Results - Discussion - Methods - Data availability section - Acknowledgements - Disclosure and Competing Interests Statement - References - Figure legends - Expanded View Figure legends

I look forward to seeing a revised form of your manuscript when it is ready.

Yours sincerely,

Referee #1:

In this manuscript Polikanov, Vázquez Laslop, Mankin, and coworkers report a comprehensive study wherein genome mining strategies were used to identify a number of novel analogues of Apidaecin 1b (Api), the prototypical type II proline rich antimicrobial peptide. This revealed a number of novel Api-like peptides with a diversity of sequences and varying antibacterial activities. Like Api, these new peptides were also found to operate via a mechanism involving arresting of the bacterial ribosome.

Comprehensive biochemical and structural studies subsequently showed that these Api analogues contain a conserved C terminal segment that interacts with the ribosome and represents the key pharmacophore for these peptides. Given the need for new antibacterial agents to address the growing problem of antibiotic resistance, the insights in this manuscript are likely to be of interest to a wide readership.

Referee #2:

In this manuscript, Huang et al. identify many new peptide antibiotics related to apidaecin (Api) in insect genomes, test their antibacterial properties, and show that many of them share a mechanism of action with Api, namely trapping ribosomes at stop codons after the nascent peptide has been released. These peptides share a conserved C-terminal sequence; x-ray structures of one of these novel peptides reveal that they bind in the ribosomal exit tunnel in more or less the same way as Api. Some of the novel peptides have better antimicrobial properties than Api, although the reason for this is not clear. The biochemical experiments and structural data are compelling, and overall, this work offers valuable new information about how peptides like these could be further developed into antimicrobials.

I have a few questions how about their findings fit into what is known about this class of antibiotics:

1. Why do PrAMPs appear to be limited to these families of insects. Is that a reflection of the bioinformatics of finding similar sequences? Or is it a unique mechanism for these insects?
2. What do we know about their likely microbial targets and resistance?

3. What is the basis of the superior antibiotic activity of these novel peptides? Additional work to show that the higher toxicity is due to better uptake or better ribosome binding would strengthen the paper. This suggestion is up to the authors and editor, I do not mean to impose it as a requirement.
4. What are the requirements for import of Api like peptides by the SbmA transporter?
5. As referred to in the introduction, previous studies have introduced point mutants into Api and screened libraries to test the effects on Api activity. Can the authors tie their results here back to any of these previous studies? In particular, what can we learn about the N-terminal sequences that are less conserved than the C-terminal pharmacophore?
6. Fig 4: Btr2 and Afl1 appear to inhibit growth but have little effect on promoting stop codon readthrough in the GFP reporter. Yet Btr2 appears to arrest ribosomes as stop codons in the toeprinting assays in Fig. 3. Can the authors speculate about these differences (beyond the sentence in the discussion currently)? E.g. Afl1 seems to be active in the drop test, had high MICs, intermediate phenotype on toeprinting, and did not induce read-through in the RFP-GFP assay.

Minor issues:

7. Fig 1d legend: the "conserved sequence is shaded in pink" doesn't seem to match the figure; I don't see any shading. Api137 is mentioned, but the sequence does not appear in the alignment (unless the same as wild-type Api?).
8. Fig 2 legend: some more information would be welcome here; what is the concentration of the peptide? What was the volume of the drop? Note that the drop was added after plating. A little more information would help the reader without referring to the methods section. Left plate, missing a space in *E. coli*.
9. Fig 3 third panel: what is the significance of the dotted lines surrounding the Oab1 lane?
10. Page 12, "we conclude that the N-terminal elements, while being important for antibacterial activity..." This statement would benefit from a citation because their importance is not shown here (by truncation or mutation).
11. Page 18, methods: 5 mL of culture was plated onto agar plates?
12. Table 1 - How many replicates were performed? Please comment on the color choices (binning the values). Any comment on why the MIC is much higher in the RPMI/Serum than TSB?

Referee #3:

Proline-rich antimicrobial peptides (PrAMPs) are a class of antibiotics that have garnered considerable attention due to their unique mechanism of action. Type II PrAMPs bind to the terminating ribosome with deacylated P-tRNA and RF1/2, thereby locking RF1/2 on the ribosome and preventing recycling. Extensive previous work has led to the modification of Apidaecin 1b, resulting in the derivative Api137, which features an N-terminal modification and a Q10R mutation, significantly enhancing its activity and serum stability.

This manuscript presents a comprehensive study on Type II PrAMPs from insects, focusing on their activity, structure, and diversity. The authors extend the study of Type II PrAMPs by identifying 71 new PrAMPs through genome mining, synthesizing 26 representative PrAMPs, demonstrating their antibacterial activity, and solving co-crystal structures of Fva1 and Api137 bound to the terminating ribosome. These newly identified peptides are quite similar, with an extremely conserved C-terminal part.

The data are solid and clearly presented. While this work may fall short of the significance and novelty typically expected in EMBO Reports, it represents the first systematic characterization of naturally occurring Type II PrAMPs.

Major Points:

The authors found that some identified PrAMPs, such as Fva1, exhibit greater activity than Api, as determined by MIC measurements. Interestingly, the C-terminal 12 residues of Fva1 are identical to those in the designer peptide Api137, which includes the Q10R mutation. As a result, the co-crystal structures are essentially the same. While this work reaffirms that Type II PrAMPs can bind to the 70S ribosome (using *Thermus thermophilus* ribosomes), these structures do not provide additional insights since the N-terminal parts remain invisible. In the MIC measurement, the authors did not include Api137, which is puzzling. To enhance the value of this work, I recommend that the authors clarify whether the enhanced activity is due to the N-terminal part or the Q10R mutation in the C-terminal, or at least provide some discussion on this aspect.

Minor Points:

1. When discussing previous structural studies of PrAMPs on the ribosome, the following citation should be included: "Lauer,

S.M., Reepmeyer, M., Berendes, O., Klepacki, D., Gasse, J., Gabrielli, S., Grubmüller, H., Bock, L.V., Krizsan, A., Nikolay, R., et al. (2024). Multimodal binding and inhibition of bacterial ribosomes by the antimicrobial peptides Api137 and Api88. *Nat Commun* 15, 3945. "

2. In Figure 6D, the labels for Fva1 appear to be incorrect, referring to Leu20 and Arg19.

Response to referees

Referee #1:

In this manuscript Polikanov, Vázquez Laslop, Mankin, and coworkers report a comprehensive study wherein genome mining strategies were used to identify a number of novel analogues of Apidaecin 1b (Api), the prototypical type II proline rich antimicrobial peptide. This revealed a number of novel Api-like peptides with a diversity of sequences and varying antibacterial activities. Like Api, these new peptides were also found to operate via a mechanism involving arresting of the bacterial ribosome.

Comprehensive biochemical and structural studies subsequently showed that these Api analogues contain a conserved C terminal segment that interacts with the ribosome and represents the key pharmacophore for these peptides. Given the need for new antibacterial agents to address the growing problem of antibiotic resistance, the insights in this manuscript are likely to be of interest to a wide readership.

We thank the referee for the favorable evaluation of our manuscript.
No response was required.

Referee #2:

In this manuscript, Huang et al. identify many new peptide antibiotics related to apidaecin (Api) in insect genomes, test their antibacterial properties, and show that many of them share a mechanism of action with Api, namely trapping ribosomes at stop codons after the nascent peptide has been released. These peptides share a conserved C-terminal sequence; x-ray structures of one of these novel peptides reveal that they bind in the ribosomal exit tunnel in more or less the same way as Api. Some of the novel peptides have better antimicrobial properties than Api, although the reason for this is not clear. The biochemical experiments and structural data are compelling, and overall, this work offers valuable new information about how peptides like these could be further developed into antimicrobials.

We are glad the referee found our manuscript scientifically solid and important.

I have a few questions how about their findings fit into what is known about this class of antibiotics:

1. Why do PrAMPs appear to be limited to these families of insects. Is that a reflection of the bioinformatics of finding similar sequences? Or is it a unique mechanism for these insects?

We clarified in the Results (p.6) that while bioinformatic search was done across the global genomic databases, the Api-like peptides were found only in the genomes of bees and wasps. The reasons for such narrow distribution are unknown but, as we

propose in the revised Discussion (p.13), it could be related to the pathogens that specifically infect only wasps and bees.

2. What do we know about their likely microbial targets and resistance?

We know very little about the true microbial targets of the PrAMPs. We mention this in a paragraph added in the revised Discussion (p.13).

3. What is the basis of the superior antibiotic activity of these novel peptides? Additional work to show that the higher toxicity is due to better uptake or better ribosome binding would strengthen the paper. This suggestion is up to the authors and editor, I do not mean to impose it as a requirement.

This is an important question, the answer to which, however, would require a much more detailed investigation of the peptide's uptake and their on-target activity. We touch upon this point in Discussion when comparing Api with a more active Fva1. Specifically, we write on p. 13: "...we favor the view that the improved activity of Fva1 relative to that of Api results from more favorable interactions with the SbmA transporter primarily responsible for delivering Api-like PrAMPs into the bacterial cytoplasm, rather than from increased affinity of the PrAMP to the ribosome".

4. What are the requirements for import of Api like peptides by the SbmA transporter?

Neither specificity, nor for that matter, physiological functions of the SbmA transporter are currently known. We indicate on p. 15 of the revised Discussion that comparing the antibacterial and on-target activities of the variety of the newly identified PrAMPs may help to illuminate SbmA characteristics.

5. As referred to in the introduction, previous studies have introduced point mutants into Api and screened libraries to test the effects on Api activity. Can the authors tie their results here back to any of these previous studies? In particular, what can we learn about the N-terminal sequences that are less conserved than the C-terminal pharmacophore?

The main conclusion from comparing our findings with the results of the previous studies is that conservation of the C-terminal pharmacophore is critical for the PrAMPs activity whereas many variations at the N-terminus are allowed. We emphasize this point in several sections of the manuscript. We further highlight in Discussion the correlation between the activity of the PrAMPs with the length and net charge of the N-terminus. We hypothesize further on p. 15 of the revised manuscript that these features likely affect the peptide's uptake.

6. Fig 4: Btr2 and Afl1 appear to inhibit growth but have little effect on promoting stop codon readthrough in the GFP reporter. Yet Btr2 appears to arrest ribosomes as stop codons in the toeprinting assays in Fig. 3. Can the authors speculate about these differences (beyond the sentence in the discussion currently)? E.g. Afl1 seems to be

active in the drop test, had high MICs, intermediate phenotype on toeprinting, and did not induce read-through in the RFP-GFP assay.

We were also puzzled that the ability of the peptides to arrest the ribosome at stop codon and to inhibit cell growth do not always correlate with their ability to induce stop codon readthrough. We have several ideas in this regard (e.g. alternative modes of action of the peptide upon the ribosome), but those ideas are too premature to impose them upon the readers of our paper. We are pursuing several experiments to gain more insights into these effects which, hopefully, would become subject of our future publications.

Minor issues:

7. Fig 1d legend: the "conserved sequence is shaded in pink" doesn't seem to match the figure; I don't see any shading. Api137 is mentioned, but the sequence does not appear in the alignment (unless the same as wild-type Api?).

We apologize for this mistake. Shading has been added. Api137 as a 'non-native' PrAMP is not shown in the alignment and we corrected the legend accordingly.

8. Fig 2 legend: some more information would be welcome here; what is the concentration of the peptide? What was the volume of the drop? Note that the drop was added after plating. A little more information would help the reader without referring to the methods section. Left plate, missing a space in E. coli.

Figure and figure legend have been modified according to the referee's suggestions.

9. Fig 3 third panel: what is the significance of the dotted lines surrounding the Oab1 lane?

We apologize for forgetting to explain the meaning of the dotted lines in the figure. In the revised manuscript we clarify that "Dotted lines around the lane with the Oab1 peptide indicate the gel lanes that contained unrelated samples and that were removed from the image."

10. Page 12, "we conclude that the N-terminal elements, while being important for antibacterial activity..." This statement would benefit from a citation because their importance is not shown here (by truncation or mutation).

We have added the missing references.

11. Page 18, methods: 5 mL of culture was plated onto agar plates?

Not exactly. The technique we use (and describe in Materials and Methods) involves pouring (not plating!) 5 ml of culture onto an agar plate and then, after spreading, aspirating the excess of the liquid. When bacterial culture forms a very thin 'pond' over

the agar surface bacterial cells rapidly attach to the agar and remain behind after the medium is aspirated. This procedure allows for formation of a uniform lawn of cells on the plate.

12. Table 1 - How many replicates were performed? Please comment on the color choices (binning the values). Any comment on why the MIC is much higher in the RPMI/Serum than TSB?

The MIC experiments were performed in triplicate and we indicate it now in the revised manuscript. We also added the legend for the color scheme used. Finally, we do not know why some PrAMPs show dramatically different MIC in 33% TSB and the more physiologically relevant RPMI/Serum media (Belanger et al., 2020). It could be related to differential peptide stability or uptake; however, since we have no clear insights, we prefer to not comment on this in the manuscript.

Referee #3:

Proline-rich antimicrobial peptides (PrAMPs) are a class of antibiotics that have garnered considerable attention due to their unique mechanism of action. Type II PrAMPs bind to the terminating ribosome with deacylated P-tRNA and RF1/2, thereby locking RF1/2 on the ribosome and preventing recycling. Extensive previous work has led to the modification of Apidaecin 1b, resulting in the derivative Api137, which features an N-terminal modification and a Q10R mutation, significantly enhancing its activity and serum stability.

This manuscript presents a comprehensive study on Type II PrAMPs from insects, focusing on their activity, structure, and diversity. The authors extend the study of Type II PrAMPs by identifying 71 new PrAMPs through genome mining, synthesizing 26 representative PrAMPs, demonstrating their antibacterial activity, and solving co-crystal structures of Fva1 and Api137 bound to the terminating ribosome. These newly identified peptides are quite similar, with an extremely conserved C-terminal part.

The data are solid and clearly presented. While this work may fall short of the significance and novelty typically expected in **EMBO** Reports, it represents the first systematic characterization of naturally occurring Type II PrAMPs.

Major Points:

The authors found that some identified PrAMPs, such as Fva1, exhibit greater activity than Api, as determined by MIC measurements. Interestingly, the C-terminal 12 residues of Fva1 are identical to those in the designer peptide Api137, which includes the Q10R mutation. As a result, the co-crystal structures are essentially the same. While this work reaffirms that Type II PrAMPs can bind to the 70S ribosome (using *Thermus thermophilus* ribosomes), these structures do not provide additional insights since the N-terminal parts remain invisible. In the MIC measurement, the authors did not include Api137, which is puzzling. To enhance the value of this work, I recommend that the

authors clarify whether the enhanced activity is due to the N-terminal part or the Q10R mutation in the C-terminal, or at least provide some discussion on this aspect.

We thank the referee for the insightful comments.

Regarding MIC testing, because this manuscript deals only with the natural PrAMPs encoded in the insect genomes, we intentionally excluded Api137 from our MIC table. However, of note, that activity of wt Fva1 and few other peptides was on par with the antibacterial activity of Api137.

Regarding structural studies, to partly address some of the referee's questions, in the revised manuscript we added one more structure, that of wt Api in complex with the *Thermus thermophilus* ribosome that allows a direct comparison of two native PrAMPs, Api and Fva1. We believe one of the important conclusions that could be drawn from all the newly determined high-resolution X-ray structures is that N-terminal variations have little effect upon placement of the C-terminal pharmacophore. This conclusion is not trivial, because Fva1 has a notably stronger antibacterial properties that, hypothetically, could result from a different mode of target engagement. Furthermore, including 'non-natural' Api137 in the structural studies made it possible to directly compare interactions of the same PrAMP with ribosomes from different species (and to show that they remain essentially unchanged).

On p. 11 of the revised manuscript, we also comment on the possible role of the Q10R substitution writing: "...neither Gln10 of Api nor Arg10 of Api137 [or of Fva1] is well resolved in the obtained electron density maps, suggesting that these residues are flexible and are unlikely to interact strongly with the ribosome or contribute to the on-target activity of these peptides."

Minor Points:

1. When discussing previous structural studies of PrAMPs on the ribosome, the following citation should be included: "Lauer, S.M., Reepmeyer, M., Berendes, O., Klepacki, D., Gasse, J., Gabrielli, S., Grubmüller, H., Bock, L.V., Krizsan, A., Nikolay, R., et al. (2024). Multimodal binding and inhibition of bacterial ribosomes by the antimicrobial peptides Api137 and Api88. *Nat Commun* 15, 3945. "

The reference was added.

2. In Figure 6D, the labels for Fva1 appear to be incorrect, referring to Leu20 and Arg19.

We thank the referee for noting this mistake, which we have now corrected.

Dear Dr. Mankin,

Thank you for the submission of your revised manuscript to EMBO Reports. As my colleague Achim Breiling is currently traveling, I have temporarily taken over the handling of your manuscript. As you will see from the reports copied below, the referees are very positive about the study and support publication.

Before we can officially accept your manuscript, I kindly ask you to address a few editorial requests, listed below.

- Please reduce the number of keywords to 5.

- Please update the 'Conflict of interest' paragraph to our new 'Disclosure and competing interests statement'. For more information see

<https://www.embopress.org/page/journal/14693178/authorguide#conflictsofinterest>

- Regarding the Author Contributions, we now use CRediT to specify the contributions of each author in the journal submission system. Therefore, please remove the Author Contributions from the manuscript file and make sure that the author contributions in our online manuscript tracking system are correct and up-to-date. The information you specified in the system will be automatically retrieved and typeset into the article. You can enter additional information in the free text box provided, if you wish.

- Please note that we encourage Data Citations, i.e., a citation of not only the manuscript but also the dataset itself. You could e.g., use this format for 6YSS (Chan et al., 2020) or for any other instances, where you reused a dataset. You first cite the paper and then in addition the dataset as follows:

In the main text, data citations are formatted as follows: "Data ref: Smith et al, 2001" or "Data ref: NCBI Sequence Read Archive PRJNA342805, 2017". In the Reference list, data citations must be labeled with "[DATASET]". A data reference must provide the database name, accession number/identifiers and a resolvable link to the landing page from which the data can be accessed at the end of the reference. Further instructions are available at

<<https://www.embopress.org/page/journal/14693178/authorguide#referencesformat>>.

- Please ensure that all information on funding that is listed in the Acknowledgments is also entered in the online manuscript tracking system. The information in the manuscript and that in the system must be congruent.

- Please provide a callout for Fig. 5A in the text.

- Please remove the Reagents and Tools table from the manuscript text and upload it as separate file (file type: Reagents table).

- Would Figure EV1 not benefit from being converted to a Table (Table EV#)?

- Table 1 and 2 should be placed between main and EV figure legends.

- Data availability section: Please provide URLs that resolve directly to the datasets at the RCSB Protein Data Bank.

- Finally, EMBO Reports papers are accompanied online by

A) a short (1-2 sentences) summary of the findings and their significance,

B) 2-3 bullet points highlighting key results and

C) a schematic summary figure that provides a sketch of the major findings (not a data image).

Please provide the summary figure as a separate file in PNG or JPG format at a size of 550x300-600 pixels (width x height).

Please note that the size is rather small and that text needs to be readable at the final size. Please send us this information along with the revised manuscript.

- On a different note, I would like to alert you that EMBO Press offers a new format for a video-synopsis of work published with us, which essentially is a short, author-generated film explaining the core findings in hand drawings, and, as we believe, can be very useful to increase visibility of the work. This has proven to offer a nice opportunity for exposure i.p. for the first author(s) of the study. Please see the following link for representative examples and their integration into the article web page:

<https://www.embopress.org/doi/full/10.15252/emj.2019103932>

With kind regards,

Referee #2:

The authors responded thoughtfully to the questions raised in the previous round of review. I have nothing further to add.

Referee #3:

The authors have answered all my concerns. The revised manuscript can be published in EMBO Reports.

All editorial and formatting issues were resolved by the authors.

Alexander Mankin
University of Illinois
Pharmaceutical Sciences
M/C 870, Rm. 3052
900 S. Ashland Ave.
Chicago, IL 60607
United States

Dear Dr. Mankin,

I am very pleased to accept your manuscript for publication in the next available issue of EMBO reports. Thank you for your contribution to our journal.

Yours sincerely,
